# Structures of G-protein coupled receptor HCAR1 in complex with Gi1 protein reveal the mechanistic basis for ligand recognition and agonist selectivity

Xin Pan[1]☯, Fang Ye[2]☯, Peiruo Ning[2]☯, Yiping Yu[3]☯, Zhiyi Zhang[2], Jingxuan Wang[2], Geng Chen[2], Zhangsong Wu[2], Chen Qiu[2], Jiancheng Li[4], Bangning Chen[5], Lizhe Zhu[3]*, Chungen Qian[5]*, Kaizheng Gong[1]*, Yang Du[2,6]*

1 Department of Cardiology, Institute of Cardiovascular Disease, Yangzhou Key Lab of Innovation Frontiers in Cardiovascular Disease, Affiliated Hospital of Yangzhou University, Yangzhou University, Yangzhou, China, 2 Kobilka Institute of Innovative Drug Discovery, The Second Affiliated Hospital, Shenzhen Futian Biomedical Innovation R&D Center, School of Medicine, Chinese University of Hong Kong, Shenzhen, China, 3 Warshel Institute for Computational Biology, School of Medicine, the Chinese University of Hong Kong, Shenzhen, China, 4 Instrumental Analysis Center, Shenzhen University, Shenzhen, China, 5 Department of Reagent Research and Development, Shenzhen YHLO Biotech Co.,Ltd., Shenzhen, China, 6 Department of Endocrinology, Peking Union Medical College Hospital, Beijing, China

☯ These authors contributed equally to this work.
* yangdu@cuhk.edu.cn (YD); yungkzh@163.com (KG); chungen_qian@hust.edu.cn (CQ); zhulizhe@cuhk.edu.cn (LZ)

## Abstract

Hydroxycarboxylic acid receptor 1 (HCAR1), also known as lactate receptor or GPR81, is a class A G-protein-coupled receptor with key roles in regulating lipid metabolism, neuroprotection, angiogenesis, cardiovascular function, and inflammatory response in humans. HCAR1 is highly expressed in numerous types of cancer cells, where it participates in controlling cancer cell metabolism and defense mechanisms, rendering it an appealing target for cancer therapy. However, the molecular basis of HCAR1-mediated signaling remains poorly understood. Here, we report four cryo-EM structures of human HCAR1 and HCAR2 in complex with the Gi1 protein, in which HCAR1 binds to the subtype-specific agonist CHBA (3.16 Å) and apo form (3.36 Å), and HCAR2 binds to the subtype-specific agonists MK-1903 (2.68 Å) and SCH900271 (3.06 Å). Combined with mutagenesis and cellular functional assays, we elucidate the mechanisms underlying ligand recognition, receptor activation, and G protein coupling of HCAR1. More importantly, the key residues that determine ligand selectivity between HCAR1 and HCAR2 are clarified. On this basis, we further summarize the structural features of agonists that match the orthosteric pockets of HCAR1 and HCAR2. These structural insights are anticipated to greatly accelerate the development of novel HCAR1-targeted drugs, offering a promising avenue for the treatment of various diseases.

**Data availability statement:** All relevant data are within the paper and its Supporting Information files. The 3D cryo-EM density maps of the CHBA-bound HCAR1-Gi1-scFV16, apo HCAR1-Gi1-scFV16, and MK-1903-, SCH900271-bound HCAR2-Gi1-scFV16 complexes have been deposited in the Electron Microscopy Data Bank database under accession codes EMD-61029, EMD-61249, EMD-61028, and EMD-61027, respectively. The atomic coordinates for the atomic models of the CHBA-bound HCAR1-Gi1-scFV16, apo HCAR1-Gi1-scFV16, and MK-1903-, SCH900271-bound HCAR2-Gi1-scFV16 complexes generated in this study have been deposited in the Protein Data Bank database under accession codes 9IZD, 9J8Z, 9IZC, and 9IZA, respectively.

**Funding:** This work was supported by the National Natural Science Foundation of China (grant number: 32271263 (Y.D.), 82470258 (X.P.)), Shenzhen Sci. & Tech Innovation Bureau JCYJ 20220818103009018 (Y.D.) and JCYJ 20240813113521028 (Y.D.), the Shenzhen-Hong Kong Cooperation Zone for Technology and Innovation HZQB-KCZYB-2020056 (Y.D.), the Natural Science Foundation of Jiangsu Higher Education Institutions of China 24KJB320025 (X.P.), the Health-related Projects of Yangzhou Basic Research Program (Joint Special Project, 2024-3-13 (X.P.)), the Jiangsu Provincial Medical Key Discipline Cultivation Unit JSDW202251 (K.G.), the Yangzhou Medical Key Discipline Cultivation YZYXZDXK-09 (K.G.), and Yangzhou Basic Research Program (Joint Special Project, 2024-01-04 (K.G.)). The funders had no role in study design, data collection and analysis, decision to publish, or preparation of the manuscript.

**Competing interests:** The authors have declared that no competing interests exist.

**Abbreviations:** 3-HBA, 3-hydroxybenzoic acid; AA, acetate; ABP, allosteric binding pocket; β-OHB, β-hydroxybutyrate; BA, butyrate; cAMP, cyclic adenosine monophosphate; CHBA, 3-chloro-5-hydroxybenzoic acid; CHS, cholesteryl hemisuccinate; CTF, contrast transfer function; cryo-EM, cryo-electron microscopy; DDM, dodecylmaltoside; DNGαi1, dominant-negative Gαi1; FFAR2, free fatty acid receptors 2; GDN, glycol-diosgenin; HA,

## Introduction

Under normal physiological conditions, adults produce approximately 1,500 mmol of lactate daily from diverse tissues, including muscle, brain, heart, gut, and skin [1,2]. Lactate not only serves as a crucial energy source and gluconeogenic precursor in vivo [3,4], but also functions as a pivotal signaling molecule that regulates a variety of physiological and pathological cellular processes [5]. The extracellular signaling roles of lactate are primarily mediated by the lactate-activated G-protein-coupled receptor GPR81, also known as hydroxycarboxylic acid receptor 1 (HCAR1) [6,7]. Although HCAR1 is mainly expressed in adipocytes and skeletal muscle cells [8], recent evidence has shown that its expression is abnormally elevated in numerous types of cancer cells, such as pancreatic, bladder, breast, lung, and colorectal [9,10]. Indeed, lactate-mediated HCAR1 activation plays an important role in cancer progression, including angiogenesis, immune evasion, and cell chemoresistance, through both autocrine and paracrine mechanisms [11,12]. Via the autocrine pathway, lactate increases the production of PD-L1 by activating HCAR1 in cancer cells, thereby facilitating immune evasion of the cells [13]; via the paracrine pathway, cancer cell-derived lactate activates HCAR1 in dendritic cells to suppress the presentation of MHCII on the cell surface, thus preventing the presentation of tumor-specific antigens to other immune cells [12]. Furthermore, a recent report indicates that lactate can drive tumor-induced cachexia through the HCAR1-Gi-Gβγ-RhoA/ROCK1-p38 signaling cascade [14]. Beyond its role in tumorigenesis, HCAR1 also participates in a range of physiological functions, including the regulation of lipid metabolism [7,8], wound healing [15], angiogenesis [16], neuroprotection [17,18], cardiovascular function [19], and inflammatory response [20,21]. Collectively, these findings suggest that HCAR1 is an emerging therapeutic target for a variety of diseases, especially cancers, but there are no HCAR1 agents clinically available to date.

HCAR1 belongs to the hydroxycarboxylic acid receptor family and displays high homology with subfamily receptors HCAR2 and HCAR3 [22]. Similar to HCAR2, studies have demonstrated that HCAR1 suppresses lipolysis by reducing cAMP levels in adipocytes via a Gi protein-coupled pathway [7,23]. It is well known that many HCAR2 ligands, including niacin, acipimox, and acifran, have been approved for clinical treatment of dyslipidemia, whereas their therapeutic value is limited by an uncomfortable cutaneous flushing effect, attributed to the activation of HCAR2 on Langerhans cells and keratinocytes [24,25]. In contrast, HCAR1 inhibits lipolysis without causing skin flushing, rendering it a more promising target for dyslipidemia treatment [23,26]. Additionally, it seems that the ligand binding sites of HCAR1 and HCAR2 bear a close structural resemblance, as their endogenous ligands, lactate and β-hydroxybutyrate (β-OHB), are structurally analogous hydroxy monocarboxylic acids [6,27]. Intriguingly, however, the effects of HCAR1 and HCAR2 on tumor growth are diametrically opposed: HCAR1 is a tumor promoter, while HCAR2 is a tumor suppressor [28–30]. Therefore, developing a single drug that can simultaneously block HCAR1 and activate HCAR2 would be ideal for cancer therapy.

hemagglutinin; HCAR1, hydroxycarboxylic acid receptor 1; LMNG, lauryl maltose neopentyl glycol; MD, molecular dynamics; OBP, orthosteric binding pocket; RMSD, root-mean-square deviation; Sf9, *Spodoptera frugiperda 9*; TM, transmembrane; WT, wild-type.

Recently, several cryo-electron microscopy (cryo-EM) structures of HCAR2 bound to niacin, acipimox, acifran, GSK256073, and MK-6892 have been successively reported, providing valuable insights into the molecular mechanisms of HCAR2 [31]–[37]. However, no experimental structures have yet been resolved for HCAR1. The mechanisms of ligand recognition, selectivity, and receptor activation in HCAR1 remain poorly understood, severely impeding the drug development process targeting HCAR1.

In this study, single-particle cryo-EM was used to determine four structures of human HCAR1 and HCAR2 in complex with the heterotrimeric Gi1 protein: HCAR1 bound to the subtype-specific agonist 3-chloro-5-hydroxybenzoic acid (CHBA) [38], HCAR1 in the absence of an agonist (apo) state, and HCAR2 bound to the subtype-specific agonists MK-1903 and SCH900271 [39,40]. Together with mutagenesis and cellular functional assays, our study revealed the ligand recognition, receptor activation, and G protein coupling mechanisms of HCAR1. In addition, the mechanism underlying ligand selectivity between HCAR1 and HCAR2 was systematically elaborated. We believe that these structural insights will significantly expedite the development of novel HCAR1-targeted drugs.

## Results

### Overall structure of HCAR1-Gi1 complex

To elucidate the molecular mechanisms of HCAR1, a stable HCAR1-Gi1 complex was prepared through co-expression of the HCAR1 receptor and Gi1 protein in Sf9 insect cells. Subsequently, the HCAR1-Gi1 complex was assembled with scFv16, a Gi1-stabilizing antibody, in the presence or absence of an agonist, obtaining the cryo-EM density maps of two HCAR1-Gi1-scFv16 complexes, bound to subtype-specific agonist CHBA (3.16 Å) and apo form (3.36 Å) (Fig 1A and 1B). Based on the high-quality density maps, we successfully built atomic models comprising the receptor HCAR1, ligand, Gi1 protein, and scFv16, in which most of the side chains were well defined (S1 and S2 Figs).

As is the case with most class A GPCRs, the overall structure of HCAR1 consisted of a canonical seven-transmembrane (TM) helical domain, whereas its extracellular conformation formed a distinct "lid-like" structure that almost completely isolated the orthosteric ligand from the external solvent (Fig 2A): (1) ECL2 was deeply inserted into the orthosteric pocket. (2) ECL1 and ECL3 closely clamped ECL2 on both sides. (3) N-terminus compressed ECL2 from the top. Further analysis revealed that the presence of three disulfide bonds ($C6^{N-term}$–$C157^{ECL2}$, $C7^{N-term}$-$C252^{ECL3}$, $C88^{3.25}$–$C165^{45.50}$) was important for maintaining the stability of the HCAR1 extracellular architecture (Fig 2B). Of these, the disulfide bond between ECL2 and TM3 ($Cys^{45.50}$–$Cys^{3.25}$) was conserved in most class A GPCRs [41], while two other disulfide bonds between the N-terminus and ECL2 ($C6^{N-term}$–$C157^{ECL2}$) and ECL3 ($C7^{N-term}$–$C252^{ECL3}$) were unique to HCAR1. Consistently, the mutations of Cys residue within these three disulfide bonds compromised the activation of HCAR1, while the expression levels were comparable to that of the wild-type (Figs 2F and S3).

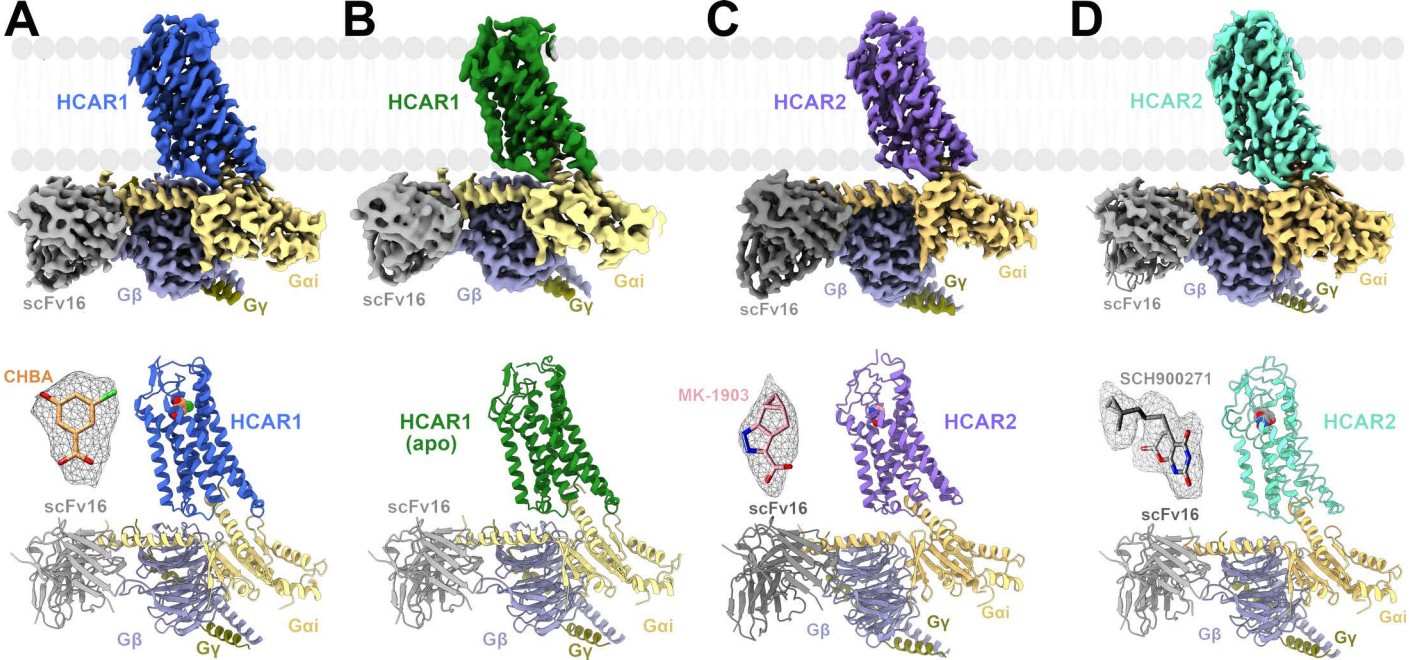

**Fig 1. Cryo-EM structures of HCAR1-Gi1 and HCAR2-Gi1 complexes. (A–D)**. Cryo-EM map and structural model of HCAR1-Gi1-scFv16 complex in the presence of CHBA **(A)** and apo form **(B)**. Cryo-EM maps and structural models of HCAR2-Gi1-scFv16 complex in the presence of MK-1903 **(C)** and SCH900271 **(D)**. The density of the agonist (shown as sticks) is depicted as gray meshes. The maps and structural models are colored by subunits. Royal blue, CHBA-HCAR1; forest green, apo-HCAR1; medium slate blue, MK-1903-HCAR2; aquamarine, SCH900271-HCAR2; light yellow, Gαi; slate blue, Gβ; olive, Gγ; dim gray, scFv16; orange, CHBA; pink, MK-1903; gray, SCH900271.

Notably, the extracellular conformation of HCAR1 shared a high topological similarity to that of the subfamily receptors HCAR2 and HCAR3 (S4A Fig); nonetheless, distinct differences were still discernible. For example, HCAR1 had a much shorter N-terminus than HCAR2 and HCAR3. The amino acid sequence alignment showed that the N-terminal lengths of the three receptors differed by 12 amino acids (S5 Fig). Moreover, the HCAR family receptors all contained three disulfide bonds; two of them ($Cys^{45.50}$–$Cys^{3.25}$–$Cys^{N-term}$–$Cys^{ECL3}$)were conserved, and one ($Cys^{N-term}$–$Cys^{ECL2}$) displayed a marked difference in the spatial position (S4A Fig). In HCAR1, the N-terminal residue $C6^{N-term}$ formed a disulfide bond with $C157^{ECL2}$ at the head of ECL2, while in HCAR2 and HCAR3, the allelic residue $C18^{N-term}$ paired with $C183^{5.33}$ at the tail of ECL2. Likely, because of the influence of these different disulfide bond connections, the N-terminus and ECL3 positions in HCAR1 were apparently shifted relative to those in HCAR2 and HCAR3 (S4B Fig).

### Ligand recognition mechanism of HCAR1

The orthosteric binding pocket (OBP) of HCAR1, positioned approximately 15 Å (measured from $C7^{N-term}$ to the ligand) within the receptor core, primarily comprised TM1, TM2, TM3, TM7, and ECL2 (Fig 2C). The subtype-specific agonist CHBA was stabilized within the OBP through an extensive network of polar and hydrophobic interactions. A detailed structural analysis showed that the carboxyl group of CHBA formed a salt bridge with $R99^{3.36}$ and a hydrogen bond with $Y268^{7.43}$ (Figs 2D and S6A). Of note, an alanine substitution of $R99^{3.36}$ in HCAR1 virtually abolished the receptor activation mediated by CHBA, implying the importance of this residue (Fig 2G). Besides, the hydroxyl group of CHBA formed two potential hydrogen bonds with $E166^{45.51}$ and $R71^{2.60}$, and the imidazole ring of $H261^{7.36}$ pressed against CHBA from the top (Fig 2D and 2E). Such a binding mode facilitated the formation of an "ionic-lock" between the acidic residue $E166^{45.51}$ and basic residues $R71^{2.60}$ and $H261^{7.36}$, further helping to maintain the ligand in a stable pose. Consistently, the mutations of

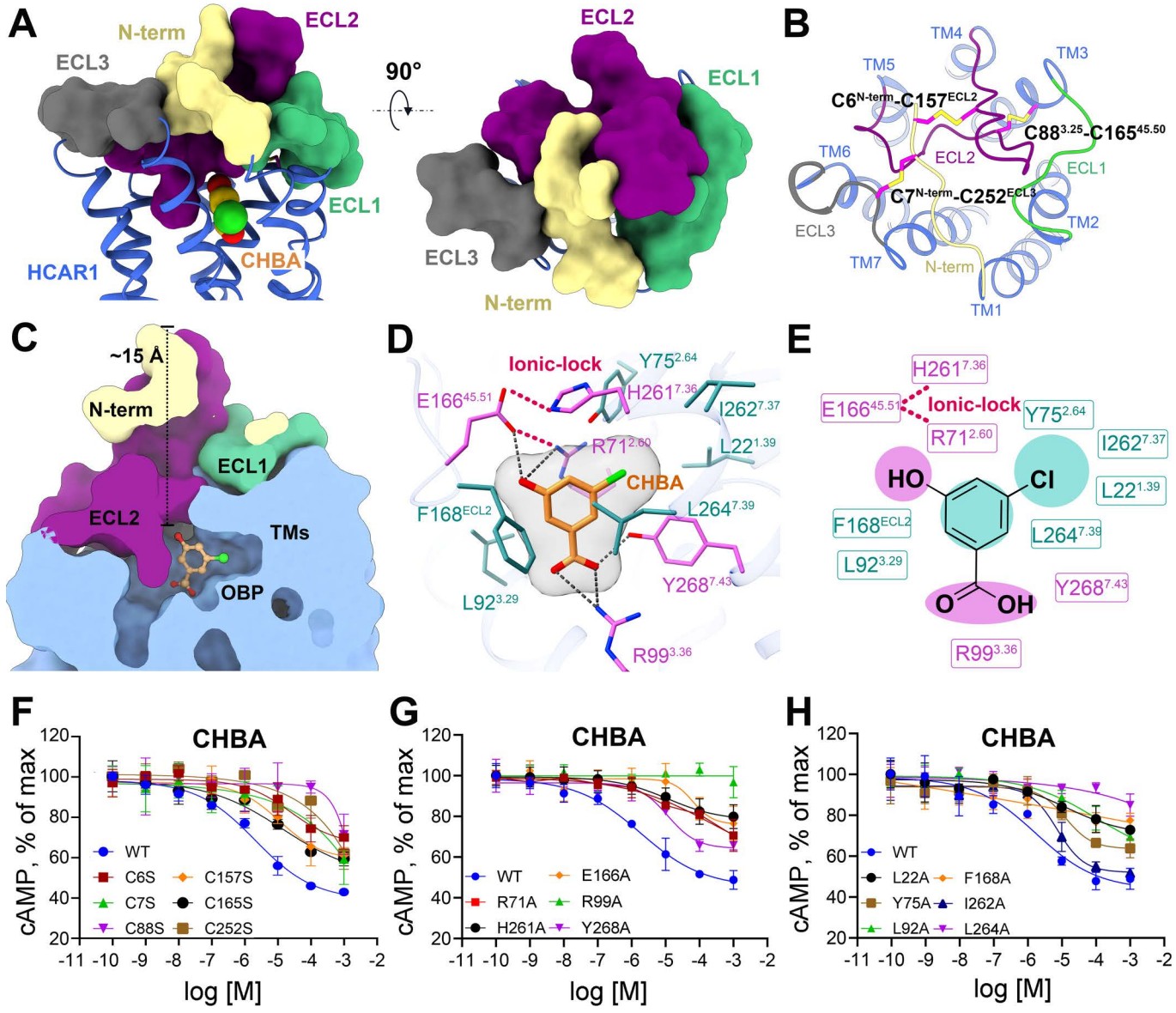

**Fig 2. Overall structure and orthosteric binding pocket of CHBA-bound HCAR1. (A)**. Extracellular architecture of CHBA-HCAR1 from side and top views. **(B)**. Three disulfide bonds (magenta sticks) are shown in the extracellular region of HCAR1. **(C)**. Vertical cross-sections of the binding pockets of CHBA in HCAR1. **(D)**. Detailed polar and hydrophobic interactions of CHBA with HCAR1. **(E)**. 2D schematic interactions of CHBA with surrounding residues. The structures of HCAR1 and agonist are colored differently. N-terminal loop (khaki), ECL1 (pale green), ECL2 (purple), and ECL3 (dark gray) are shown as surface representations. Key hydrophilic residues (orchid), hydrophobic residues (turquoise), and CHBA (orange) are shown as sticks. Black dashed line, polar interactions; violet red dashed line, ionic-lock. **(F)**. Effects on Gi-mediated cAMP by single-point mutations of C6[N-term], C157[ECL2], C7[N-term], C252[ECL3], C88[3.25], and C165[45.50] that disrupt the disulfide bonds. **(G, H)**. Effects on Gi-mediated cAMP by single point mutations of key hydrophilic and hydrophobic residues that interact with CHBA. The data are presented as means ± SEM. The experiments are performed in triplicate. The underlying data can be found in S1 Data.

E166[45.51], R71[2.60] and H261[7.36] to alanine greatly impaired the agonistic activity of CHBA (Fig 2G). In addition to the polar interactions, both the aromatic ring and chloro group of CHBA engaged in robust hydrophobic interactions with several surrounding hydrophobic residues, including L22[1.39], Y75[2.64], L92[3.29], F168[ECL2], I262[7.37], and L264[7.39] (Figs 2D and S6A). The cAMP accumulation assay revealed that these hydrophobic residue mutations, particularly F168[ECL2] and L264[7.39], led

to a noticeable reduction in agonistic activity, highlighting the necessity of the OBP hydrophobic environment for HCAR1 activation by CHBA (Fig 2H).

Previous structure-activity relationship studies underscored the significance of the substituent at the 5-position of CHBA in its agonistic activity on HCAR1 [20,38,42]. Our structural snapshots provided a mechanistic rationale for this observation. Regarding the hydrophobicity and hydrophilicity of the 5-position substituent, a hydrophobic group (e.g., methyl, fluoro, chloro, and bromo) appeared to have greater agonist activity than a hydrophilic group (e.g., hydroxyl), as evidenced by their respective $EC_{50}$ values (S7A Fig) [38]. Our structural analysis suggested that this preference was mainly attributed to the hydrophobic microenvironment surrounding the 5-position of CHBA, which was encircled by a series of hydrophobic residues, $L22^{2.60}$, $Y75^{2.64}$, $I262^{7.37}$, and $L264^{7.39}$ (S7B Fig). With respect to the size of the substituent, a larger group at the 5-position of CHBA (e.g., tert-butyl, trifluoromethyl, methoxy, and phenyl) resulted in a dramatic loss of agonistic activity [38]. This was likely due to severe steric hindrance between these larger substituents and the surrounding $Y75^{2.64}$, $H261^{7.36}$, and $I262^{7.37}$ (S7C–7E Fig). Together, our findings elucidated the ligand recognition mechanism of HCAR1, providing critical guidance for the rational design of novel HCAR1-targeted drugs.

## Ligand selectivity between HCAR1 and HCAR2

As members of the same receptor family, HCAR1 and HCAR2 share up to 52% sequence identity [22], and their overall structures are very similar as well, with root-mean-square deviation (RMSD) values of 0.8 Å for the Cα atoms. However, in fact, there is a marked disparity in their ligand selectivity. For example, the subtype-specific agonist CHBA selectively activates HCAR1 without affecting HCAR2 [38]. Conversely, several HCAR2 agonists, such as acipimox and acifran, fail to elicit agonistic responses toward HCAR1 [7,8,43]. Moreover, recent findings have identified an allosteric binding pocket (ABP) in HCAR2, which can be activated by the allosteric agonist compound 9n [31]. Nevertheless, compound 9n appears to exert no allosteric activity toward HCAR1. These previous observations prompted us to question as to what the structural differences determine the ligand selectivity for HCAR1 and HCAR2.

When focusing on the OBP regions of HCAR1 and HCAR2, we noted that their TM1, TM2, TM3, and TM7 were highly superimposed (Fig 3B). Sequence alignment showed that most of the residues involved in forming the OBP were conserved in HCAR1 and HCAR2 (Fig 3A). In particular, the positively charged residue $R99/R111^{3.36}$, considered crucial for the recognition of the agonist carboxyl group, occupied an almost identical position in both receptors (Fig 3C).

Despite this, several distinct differences were still observed in the OBP regions of HCAR1 and HCAR2. (1) The ECL2 regions of the two receptors showed marked differences in spatial position (Fig 3B). Particularly, the Cα atom of $S167^{45.52}$ in HCAR1 moved about 3.3 Å relative to $S179^{45.52}$ in HCAR2 (Fig 3C). Meanwhile, the key residue $E166^{45.51}$ in the ECL2 of HCAR1, which interacted with CHBA, was replaced by $S178^{45.51}$ in HCAR2. To explore the role of ECL2 in agonist selectivity between HCAR1 and HCAR2, molecular dynamics (MD) simulations were conducted (S8A and S8B Fig). After performing 200 ns of MD simulations on CHBA-bound HCAR1 and MK1903-bound HCAR2, it was found that the RMSD values of the ECL2 regions for both complexes exceeded 10 Å, which indicated a high degree of flexibility in this loop region (S8C–8F Fig). Furthermore, interaction analysis revealed that CHBA primarily formed hydrogen bonds with the carboxyl group of $E166^{45.51}$ in HCAR1 (S8G Fig). In contrast, MK1903 formed stable hydrogen bonds with the side chain and backbone of $S179^{45.52}$ in HCAR2 (S8H Fig). We considered that the differences in the interactions between ligands and ECL2 might be a crucial factor influencing agonist selectivity between HCAR1 and HCAR2. (2) The residue $R79^{ECL1}$ in HCAR1 was replaced by the bulkier residue $W91^{ECL1}$ in HCAR2. This substitution led to $Y87^{2.64}$ in HCAR2 shifting closer to the chloro group of CHBA relative to $Y75^{2.64}$ in HCAR1, thus causing a steric clash (Figs 3C and S9A). Notably, when $W91^{ECL1}$ in HCAR2 was mutated to Arg, CHBA displayed weak agonistic activity toward HCAR2 (S9B Fig). Conversely, mutating $R79^{ECL1}$ in HCAR1 to Trp significantly impaired the downstream signaling pathway activated by CHBA. Based on the above comprehensive analysis, our results suggested that the steric hindrance between the 5-position substituent of CHBA and the side chain of $Tyr^{2.64}$ was the decisive factor that CHBA bound to HCAR1 but not HCAR2. Likewise, this

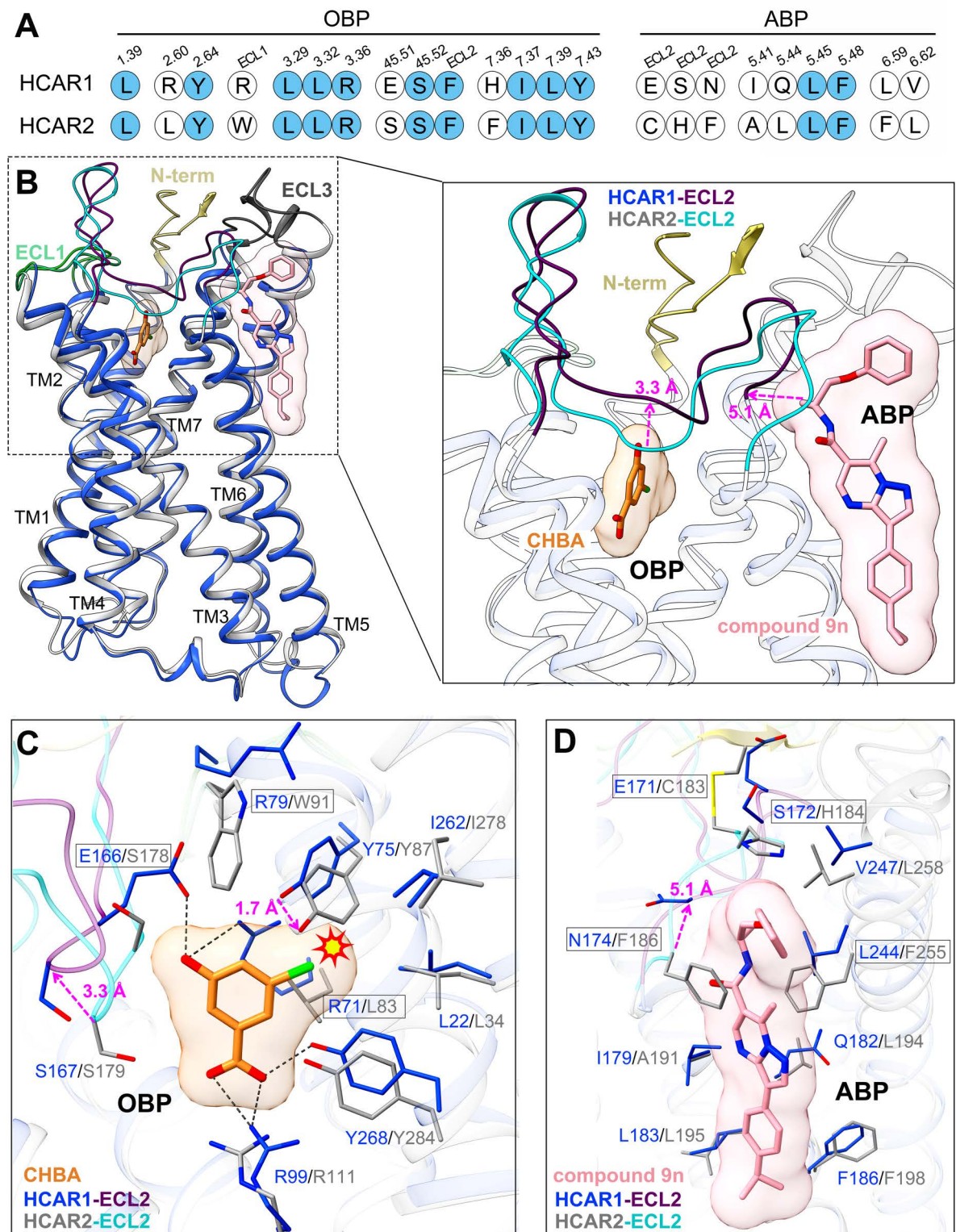

**Fig 3. Structural basis for ligand selectivity between HCAR1 and HCAR2. (A).** Sequence alignment of residues in HCAR1 and HCAR2. Conserved residues are highlighted in solid light blue circles. **(B)** Superposition of the CHBA-HCAR1 and compound 9n-HCAR2 (PDB: 8J6Q) cryo-EM structures. **(C).** Comparison of the CHBA binding modes in the OBP regions of HCAR1 and HCAR2. **(D).** Comparison of the compound 9n binding modes in the

ABP regions of HCAR1 and HCAR2. The structures of HCAR1, HCAR2, and agonists are colored differently. Royal blue, HCAR1; light gray, HCAR2; khaki, N-terminal loop; pale green, ECL1; purple, HCAR1-ECL2; cyan, HCAR2-ECL2; dark gray, ECL3; orange, CHBA; pink, compound 9n ; black dashed line, polar interactions. Key residues and agonists are shown as sticks.

finding could also explain the selectivity of many other HCAR1 agonists. For example, 3-hydroxybenzoic acid (3-HBA), which lacks a substituent at the 5-position, acted as an agonist for both HCAR1 and HCAR2; however, it showed activity exclusively toward HCRA1 but not HCRA2 when a substituent (e.g., hydroxyl, methyl, fluoro, or bromo) was introduced at the 5-position (S9D Fig) [38,42].

In the ABP region of HCAR2, the allosteric agonist compound 9n was accommodated within an amphipathic cavity composed of ECL2, TM5, and TM6 (Fig 3B). Meanwhile, compound 9n was sandwiched by the side chains of H184$^{ECL2}$, F186$^{ECL2}$ and F255$^{6.59}$ to form the aromatic stacking (Fig 3D). However, sequence alignment revealed that the residues crucial for ABP formation were poorly conserved, with the pivotal residues in HCAR2 of H184$^{ECL2}$, F186$^{ECL2}$, and F255$^{6.59}$ replaced by S172$^{ECL2}$, N174$^{ECL2}$ and L244$^{6.59}$ in HCAR1 (Fig 3A and 3D). In addition, the ECL2 segment associated with the ABP formation in HCAR1 underwent a significant conformational shift, with a maximum displacement of 5.1 Å (Fig 3B and 3D). All these structural discrepancies precluded HCAR1 from forming an ABP analogous to that observed in HCAR2, thereby explaining the lack of activity exhibited by compound 9n on HCAR1 (S9C Fig).

## Summary of the structural features of HCAR1 and HCAR2 agonists

In order to reveal the structural features of agonists that matched the HCAR1 and HCAR2 orthosteric pocket, a comprehensive comparative analysis of the binding modes of HCAR1 and HCAR2 with various agonists was imperative. Previous studies have reported numerous subtype-specific HCAR2 agonists with nanomolar potencies ($pEC_{50}$: 7.5–8.7), mainly divided into four categories: (1) anthranilic acid derivatives, such as MK-6892 [44]; (2) xanthine acid derivatives, such as GSK256073 [45]; (3) pyrazole acid derivatives, such as MK-1903 [39]; (4) thiobarbituric acid derivatives, such as SCH900271 [40] (Fig 4A). Although, the structures of HCAR2 in complex with MK-6892 and GSK256073 have been successfully determined [33,34], the molecular mechanisms by which MK-1903 and SCH900271 bind to HCAR2 remain to be fully elucidated. Particularly, SCH900271 stands out as the most potent HCAR2 agonist identified to date, with an $EC_{50}$ value as low as 2 nM [40].

In view of this, we resolved the cryo-EM structures of HCAR2 bound to MK-1903 (2.68 Å) and SCH900271 (3.06 Å) (Figs 1C, 1D, S10 and S11). The interaction analysis showed that both MK-1903 and SCH900271 established a salt bridge with R111$^{3.36}$ and hydrogen bonds with S179$^{45.52}$ and Y284$^{7.43}$, resembling the interaction patterns observed with other HCAR2 agonists (S12 Fig) [33–37]. Alanine scanning mutagenesis of R111$^{3.36}$, S179$^{45.52}$, and Y284$^{7.43}$ further corroborated the pivotal roles of these residues in ligand recognition of HCAR2 (S12B and 12D Fig). Of particular note, SCH900271 has a long hydrophobic pentyl chain modified with a cyclopropyl group that acted as a nail by inserting into a hydrophobic cavity formed by L83$^{2.60}$, W91$^{ECL1}$, M103$^{3.28}$, and L104$^{3.29}$ (S12C Fig). These extra hydrophobic interactions might explain the reason for the higher affinity of SCH900271 for HCAR2 than the other agonists. Afterwards, we analyzed the binding modes of HCAR2 with multiple agonists and found that most HCAR2 subtype-specific agonists, such as acipimox, acifran, GSK256073, MK-1903, and SCH900271, similar to the HCAR1 agonist CHBA, exclusively bound to the OBP region, whereas only MK-6892 was found to occupy two subpockets: a canonical OBP and an extended binding pocket (EBP) (S13 Fig).

We then superimposed the orthosteric agonist structures of HCAR1 and HCAR2 in OBP. For a clear comparison, the OBP was subdivided into three parts, defined as OBP1, OBP2, and OBP3 (Fig 4B). Notably, the HCAR1 agonist CHBA mainly occupied the OBP1 and OBP2 regions, whereas the majority of HCAR2 agonists were primarily localized within the OBP1 and OBP3 regions (S14F Fig). A deeper analysis indicated that the OBP3 entrance of HCAR1 was occluded by the elongated side chain of R71$^{2.60}$ (Fig 4C). Because of this, HCAR2 agonists, such as acipimox, MK-1903 (with

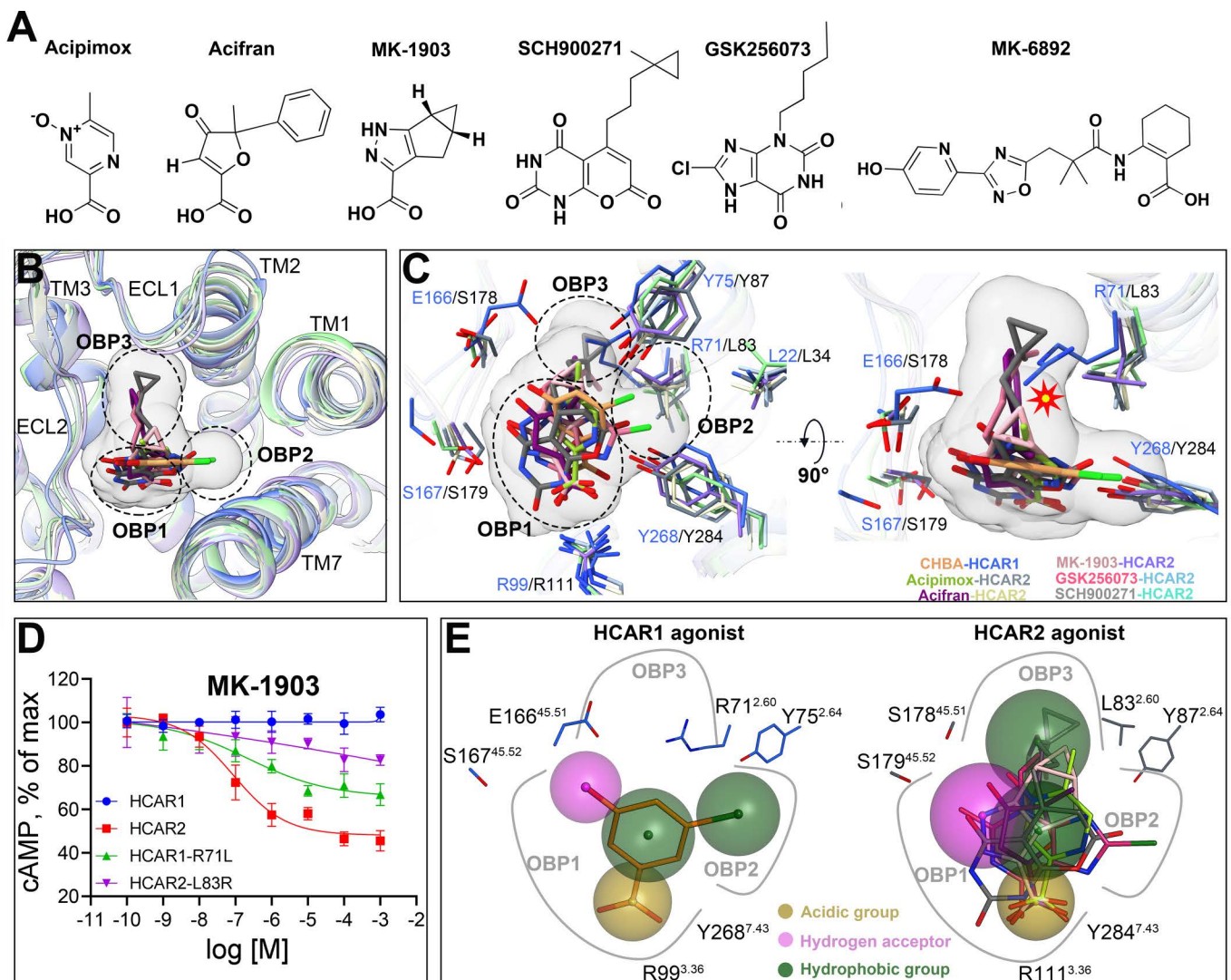

**Fig 4. Structural features of agonists that match the HCAR1 and HCAR2 orthosteric pocket. (A)**. Chemical structures of the HCAR2 subtype-specific agonists acipimox, acifran, MK-1903, SCH900271, GSK256073, and MK-6892. **(B)**. Superposition of the orthosteric agonists of HCAR1 and HCAR2, including CHBA, acipimox (PDB: 8IJB), acifran (PDB: 8IHI), GSK256073 (PDB: 8I7W), MK-1903, and SCH900271. **(C)**. Comparison of different ligand binding modes of HCAR1 and HCAR2 in the OBP regions. Orange-royal blue, CHBA-HCAR1; yellow green-slate gray, acipimox-HCAR2; pink-medium slate blue, MK-1903-HCAR2; purple-lemon chiffon, acifran-HCAR2; rose red-light sky blue, GSK256073-HCAR2; gray-aquamarine, SCH900271-HCAR2. **(D)**. Effects on Gi-mediated cAMP by single point mutations of R71$^{2.60}$L in HCAR1 and L83$^{2.60}$R in HCAR2. The data are presented as means ± SEM. The experiments are performed in triplicate. The underlying data can be found in S1 Data. **(E)**. Structural features that differentiate HCAR1 and HCAR2 agonists. Dark goldenrod, acidic group; magenta, hydrogen bond acceptor; dark green, hydrophobic group.

a small hydrophobic group), or acifran, GSK256073, SCH900271 (with a large hydrophobic group), all encountered steric hindrance with R71$^{2.60}$ in HCAR1 (S14A–14E Fig). In contrast, the allelic residue in HCAR2 was substituted with a smaller L83$^{2.60}$, creating an adequately spacious OBP3 to accommodate the functional groups of HCAR2 agonists. Our findings provided an explanation for the selective binding of HCAR2 agonists to HCAR2 rather than HCAR1. This observation was validated by the mutagenesis studies. Taking MK-1903 as an example, the R71$^{2.60}$L mutation in HCAR1 partially restored MK-1903-induced activation, whereas the L83$^{2.60}$R mutation in HCAR2 markedly impaired the agonistic activity (Fig 4D).

Given the above, we further summarized the structural features that differentiated HCAR1 and HCAR2 agonists. For HCAR1 subtype-specific agonists, the key features included an acidic moiety in OBP1 that engaged in salt bridges and hydrogen bonding with the residues R99$^{3.36}$ and Y268$^{7.43}$, a hydrogen bond acceptor in OBP1 that interacted with E166$^{45.51}$ and R71$^{2.60}$, and a hydrophobic moiety in OBP2 that participated in hydrophobic interactions (Fig 4E). The critical determinant for the specificity of HCAR1 agonist was the absence of a steric clash with Y75$^{2.64}$ in HCAR1, whereas presence with Y87$^{2.64}$ in HCAR2. For HCAR2 subtype-specific agonists, the defining characteristics were an acidic group in OBP1 that formed salt bridges and hydrogen bonds with R111$^{3.36}$ and Y284$^{7.43}$, a hydrogen bond acceptor in OBP1 that interacted with S179$^{45.52}$, and a hydrophobic moiety in OBP3 that contributed to hydrophobic interactions (Fig 4E). The specificity-determining element for HCAR2 agonist lied in the absence of a steric clash with L83$^{2.60}$ in HCAR2, while presence with R71$^{2.60}$ in HCAR1. Altogether, these insights offered a foundational framework for the rational design of highly subtype-specific agonists targeting HCAR1 and HCAR2, with the potential to enhance therapeutic precision and reduce off-target effects.

## Activation mechanism of HCAR1 receptor

The existence of the apo state of HCAR1-Gi1 complex implied that HCAR1 did not necessarily require ligand binding to activate the downstream signaling transducers, as its ECL2 might act as a built-in "agonist" (S15A Fig). In fact, a self-activated state was also observed in the subfamily receptor HCAR2 and HCAR3 (S15C Fig) [33,46]. Moreover, our signaling assay results showed that in the absence of a ligand, the downstream cAMP level decreased as the concentration of the HCAR1 receptor increased, which further corroborated the basal activity of HCAR1 (S15B Fig).

To uncover the activation mechanism of HCAR1, structural comparisons of the active HCAR1 in the apo and CHBA-bound forms with the inactive HCAR2 (PDB: 7ZLY) were performed [47]. Compared to the inactive HCAR2, both the states of HCAR1, with or without an agonist, adopted a fully active conformation, as the cytoplasmic side of TM6 showed a pronounced outward movement of 3.6 Å, which is a typical activation trait of class A GPCRs (Fig 5A). Concurrently, the extracellular side of TM5 underwent an inward movement of 4.6–5.4 Å, while the cytoplasmic side shifted outward by 2.5 Å. These conformational changes facilitated the insertion of the α5 helix of Gαi into the helical bundle of HCAR1.

With respect to the OBP region, we noted that the side chain of R99$^{3.36}$ in CHBA-bound HCAR1 stretched approximately upward 180° relative to the inactive HCAR2, thereby directly interacting with the ligand (Fig 5B). In contrast, the R99$^{3.36}$ in the apo state of HCAR1 was located in a similar position as the R111$^{3.36}$ in the inactive HCAR2, due to the absence of a ligand. Besides, in most class A GPCRs, the triggering of activation is initiated by a conserved residue W$^{6.48}$ [48]. However, the HCAR family belongs to the δ-branch of class A GPCRs, in which W$^{6.48}$ is replaced by F$^{6.48}$ or Y$^{6.48}$ at this position [36]. Our mutagenesis study showed that substituting Y233$^{6.48}$ in HCAR1 with either a larger Trp or a smaller Ala significantly impaired receptor activation, while substituting with a Phe, which has a similar size and structure as Tyr, exerted a negligible effect (Fig 5D).

Upon agonist binding, several key motifs of HCAR1, including C$^{6.47}$Y$^{6.48}$xP$^{6.50}$, P$^{5.50}$–I$^{3.40}$–F$^{6.44}$, D$^{7.49}$P$^{7.50}$xxY$^{7.53}$, and D$^{3.49}$R$^{3.50}$Y$^{3.51}$ (where x denotes any residue), underwent a series of intricate and sequential conformational rearrangements to transmit extracellular signals into the cell (S6B Fig). Specifically, in contrast to inactive HCAR2, the side chain of Y233$^{6.48}$ in HCAR1 rotated about 90°, serving as a pivotal "toggle switch" for receptor activation (Fig 5C). Immediately afterward, a pronounced structural rearrangement occurred in the triad motif P188$^{5.50}$–I103$^{3.40}$–F229$^{6.44}$, thus leading to the outward movement of TM6 (Fig 5E). As the activation signal propagated through the conserved D$^{7.49}$P$^{7.50}$xxY$^{7.53}$ motif to the bottom D$^{3.49}$R$^{3.50}$Y$^{3.51}$ motif, similar conformational changes were also observed. For example, the residues P275$^{7.50}$ and Y278$^{7.53}$ in TM7 of HCAR1 experienced rotations of 40° and 75°, respectively (Fig 5F). Moreover, in the inactive HCAR2, D124$^{3.49}$ formed a salt bridge with R125$^{3.50}$, thereby locking the receptor in an inactive state (Fig 5G). Nevertheless, in the active HCAR1, a rotameric shift of R113$^{3.50}$ disrupted the salt bridge, which contributed to the formation of an active state.

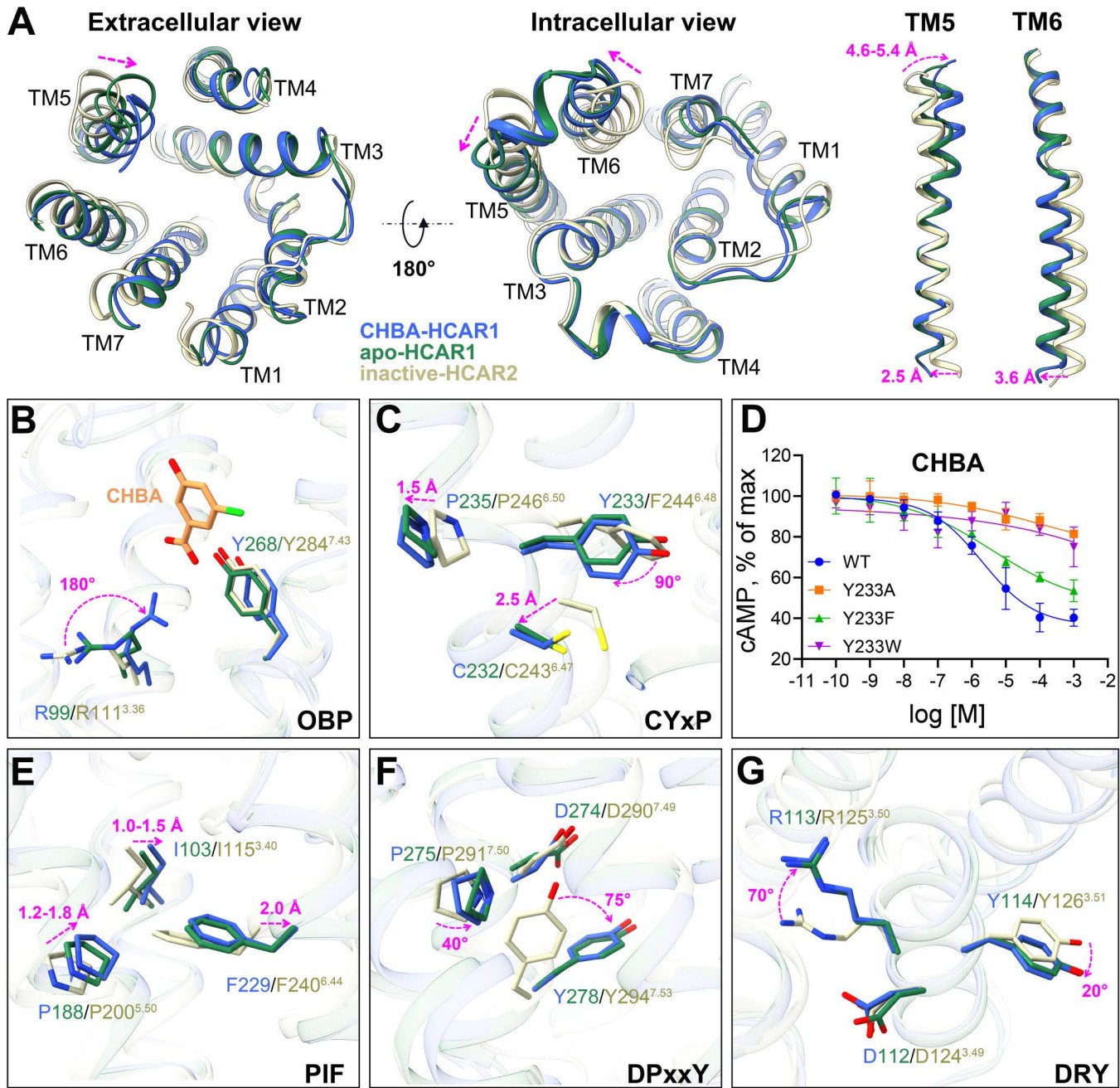

**Fig 5. Analysis of the activation mechanism of HCAR1. (A)**. Comparison of active HCAR1 with inactive HCAR2 (PDB: 7ZLY). **(B)**. Analysis of the key residues in OBP region between active HCAR1 and inactive HCAR2. **(C)**. Analysis of the differences in the $C^{6.47}Y^{6.48}xP^{6.50}$ motif. **(D)**. Effects on Gi-mediated cAMP by single point mutations of $Y233^{6.48}$ in HCAR1. The data are presented as means ± SEM. The experiments are performed in triplicate. The underlying data can be found in S1 Data. (E). Analysis of the differences in the $P^{5.50}$–$I^{3.40}$–$F^{6.44}$ motif. **(F)**. Analysis of the differences in the $D^{7.49}P^{7.50}xxY^{7.53}$ motif. **(G)**. Analysis of the differences in the $D^{3.49}R^{3.50}Y^{3.51}$ motif. Royal blue, CHBA-HCAR1; forest green, apo-HCAR1; light goldenrod yellow, inactive HCAR2; orange, CHBA; magenta arrow, shift in HCAR1 with respect to inactive HCAR2.

All these conformational rearrangements facilitated the engagement of HCAR1 with the Gi1 protein, culminating in the formation of a fully active receptor conformation.

## Interfaces between the HCAR1 receptor and Gi1

The complex structures of HCAR1-Gi1 in the apo and CHBA-bound forms exhibited almost identical G protein coupling interfaces (Fig 6A). The interactions between HCAR1 and Gi1 were primarily mediated by the α5 helix of the Gαi subunit, which inserted into the receptor cores composed of TM2, TM3, TM5, TM6, ICL2, and ICL3. It seemed that the αN helix of the Gαi subunit did not directly interact with HCAR1. As shown in Fig 6C–6F, the amphipathic C-terminus of α5 helix formed extensive polar and hydrophobic interactions within the HCAR1 cytoplasmic cavity to stabilize the conformation of HCAR1-Gi1 complex. To be specific, four hydrophobic residues of the α5 helix, namely I344, L348, L353, and F354, were embedded in a hydrophobic groove of HCAR1 constituted by residues of TM5 (I199$^{5.61}$ and L203$^{5.65}$), TM6 (M215$^{6.30}$, A218$^{6.33}$, F221$^{6.36}$, and I222$^{6.37}$), and ICL3 (L209$^{ICL3}$), thereby establishing extensive hydrophobic interactions (Figs 6C, 6E and S6C). In addition, the positively charged residue R206$^{ICL3}$ at the cytoplasmic end of ICL3 formed a salt bridge with the negatively charged residue D341 of the α5 helix. And the residue H121$^{34.51}$ at the cytoplasmic end of ICL2 formed a hydrogen bond with T340 (Fig 6D and 6F).

Subsequently, the HCAR1-Gi1 complex was structurally aligned with its subfamily receptors HCAR2 and HCAR3. The most prominent differences emerged from the positions and orientations of the α5 and αN helices (Fig 6B). Relative to the HCAR1-Gi1 complex, the α5 and αN helices underwent a clockwise rotation of approximately 3.2 and 6.9 Å in HCAR2-Gi1, and 2.8 and 8.2 Å in HCAR3-Gi1. Previous studies have emphasized the critical roles of the ICL2 and ICL3 regions in G protein coupling [33,49]. In the case of HCAR1–3 receptors, sequence analysis revealed that despite the presence of two distinct residues in the ICL2 region, their three-dimensional structures overlapped well (S16A–16F Fig). In contrast, the ICL3 region showed a clear sequence divergence: (1) HCAR1 contained one additional residue (Q208$^{ICL3}$) compared to HCAR2/3. (2) Three residues in HCAR1, namely L209$^{ICL3}$, A210$^{ICL3}$, and Q212$^{ICL3}$, were substituted with M220$^{ICL3}$, D221$^{ICL3}$, and H223$^{ICL3}$ in HCAR2/3 (S16G–16J Fig). Consequently, a notable conformational difference was observed in the ICL3 region between HCAR1 and HCAR2/3, which might explain the different shifts of the α5 and αN helices (S16B Fig). Overall, our findings clarified the Gi1 coupling features of HCAR1 and enhanced the understanding of the G protein coupling mechanism.

## Discussion

Recently, the cryo-EM structures of HCAR2 and HCAR3 bound to various agonists have been resolved [31–37,46]. However, no precise structures of HCAR1 have been reported to date. The present study reported the cryo-EM structures of HCAR1-Gi1 complex in the CHBA-bound and apo forms. As the final piece in the HCAR family puzzle, our structural insights refined the understanding of the molecular mechanisms of HCAR1. Studies of ligand recognition mechanisms suggested that the orthosteric agonist CHBA bound to HCAR1 by directly interacting with four key residues, R99$^{3.36}$, Y268$^{7.43}$, E166$^{45.51}$, and R71$^{2.60}$, in the OBP. At the same time, residues E166$^{45.51}$, R71$^{2.60}$, and H261$^{7.36}$ collectively formed an "ionic-lock" to further stabilize the binding poses of CHBA. Notably, a comparison of the apo and CHBA-bound HCAR1 structures suggested that the addition of the ligand induced a large conformational change in the side chain of R99$^{3.36}$, thus facilitating the formation of a salt bridge with the carboxyl group of CHBA (Fig 7). In fact, the positively charged residue Arg$^{3.36}$ was conserved in all three HCAR receptors and occupied a nearly identical spatial position. This explained why HCAR receptor agonists must bear an acidic functional group [50]. Furthermore, when examining the ABP region, we observed distinct structural differences between HCAR1 and HCAR2, mainly due to the presence of non-conserved key residues and a conformational shift in ECL2. Consequently, HCAR1 was unable to form an ABP region analogous to that of HCAR2.

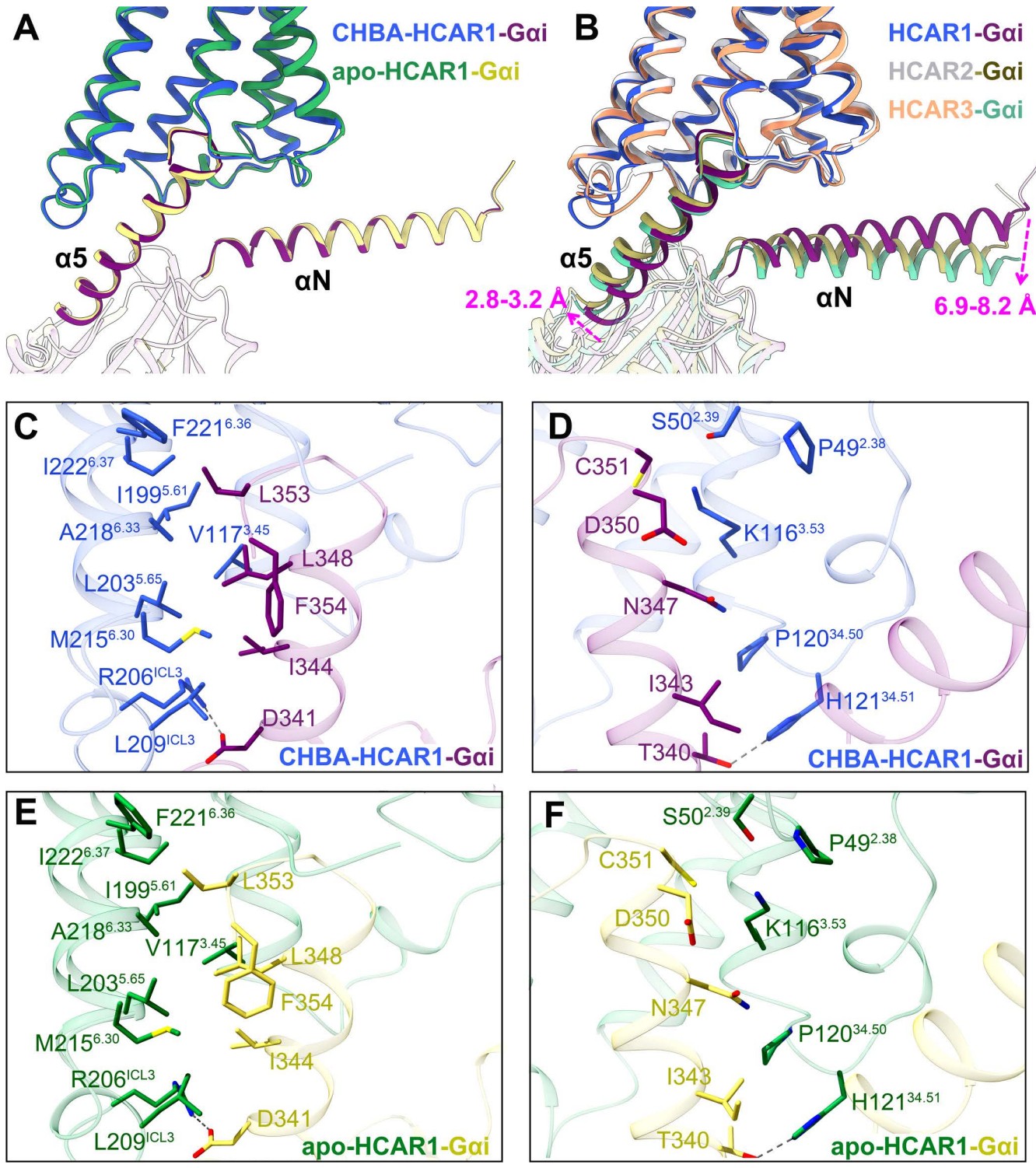

**Fig 6. Analysis of the HCAR1-Gi1 interface. (A)**. Comparison of the HCAR1-Gi1 complex with or without an agonist. **(B)**. Superimposition of the receptor G protein coupling interfaces for HCAR1-Gi1, HCAR2-Gi1 (PDB: 8J6Q), and HCAR3-Gi1 (PDB: 8IHJ). **(C−F)** Interactions of HCAR1 with the α5 helix of Gαi. Royal blue-dark magenta, CHBA-HCAR1-Gi1; forest green-beige, apo-HCAR1-Gi1; light gray-dark khaki, HCAR2-Gi1; sandy brown-medium aquamarine, HCAR3-Gi1; dark gray dashed lines, polar interactions; magenta arrow, shift with respect to HCAR1-Gi1.

PLOS Biology

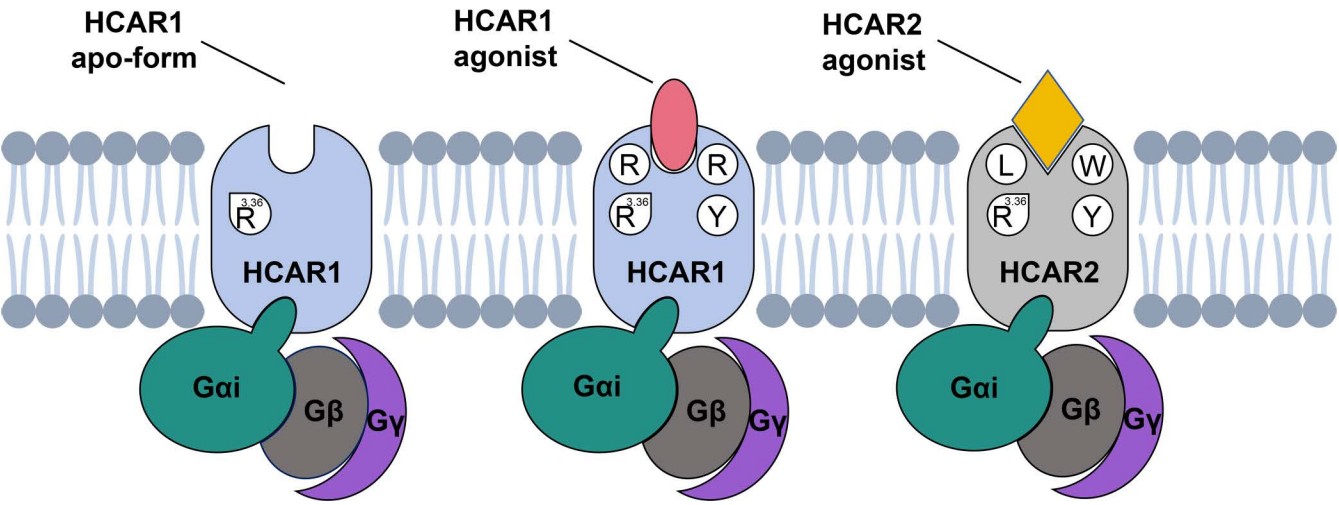

**Fig 7. Mechanism diagram of ligand recognition and selectivity of HCAR1.** This study mainly showcases the HCAR1-Gi1 complexes in both the agonist-bound and apo forms, and the HCAR2-Gi1 complexes in the agonist-bound form. Four residues of HCAR1, namely R99[3.36], Y268[7.43], E166[45.51], and R71[2.60], are deemed crucial for ligand recognition. Notably, R99[3.36] undergoes a significant conformational change upon agonist binding. Furthermore, three residues in HCAR1/HCAR2, namely R71/L83[2.60], R79/W91[ECL1], and Y75/Y87[2.64], play a decisive role in agonist selectivity between HCAR1 and HCAR2. Light blue, HCAR1; light gray, HCAR2; light coral, HCAR1 agonist; gold, HCAR2 agonist; dark cyan, Gαi; dark gray, Gβ; dark orchid, Gγ.

The HCAR family comprises three members: HCAR1, HCAR2 (GPR109A), and HCAR3 (GPR109B). Their endogenous ligands are lactate, β-OHB, and 3-hydroxyoctanoic acid, respectively [22]. Interestingly, the endogenous ligands of free fatty acid receptors 2 and 3 (FFAR2/3) are short-chain fatty acids as well [51,52]. Of these, FFAR2 is predominantly activated by acetate (AA), butyrate (BA), and propionate, while FFAR3 is activated by BA, propionate, valerate, and caproate [53]. Structural comparisons revealed that the positions of agonists in HCAR1/2 and FFAR2/3 were entirely different (S17A Fig). In HCAR1/2, the agonists CHBA and β-OHB were located within the orthosteric pocket formed by TM1, TM2, TM3, TM7, and ECL2. In FFAR2/3, the agonists AA and BA were situated in a pocket composed of TM3, TM4, TM5, TM6, and TM7. Further analysis suggested that despite the significant chemical similarity shared by the endogenous ligands of the HCAR and FFAR families, the key amino acids responsible for recognizing the acidic group of agonists were not conserved (S17B–17G Fig). Specifically, the Arg[3.36] was regarded as a critical determinant for the recognition of agonist acidic group in HCAR1/2, while the residue was substituted with Ile[3.36] in FFAR2/3 (S17E Fig). In contrast, the carboxyl group of AA and BA formed salt bridges with several basic residues Arg[5.39] His[6.55] and Arg[7.35] in FFAR2/3, whereas these residues were replaced by His[5.39], Arg[6.55] and Leu/Phe[7.35] in HCAR1/2 (S17F Fig). We speculated that these differences in key amino acids might ultimately result in distinct ligand binding modes in the HCAR and FFAR families (S17G Fig).

Regarding the mechanism of agonist selectivity, our results suggested that three residues in HCAR1/HCAR2, R79/W91[ECL1] Y75/Y87[2.64], and R71/L83[2.60], played a decisive role (Fig 7). For the HCAR1 subtype-specific agonist CHBA, the bulky residue W91[ECL1] in HCAR2 led to Y87[2.64] shifting too close to the 5-position chloro group of CHBA relative to Y75[2.64] in HCAR1, thus generating steric hindrance effect. Consequently, CHBA selectively activated HCAR1 but not HCAR2. This finding was also confirmed by the studies of Dvorak and colleagues and Liu and colleagues, in which several CHBA derivatives with a 5-position substituent exclusively bound to HCAR1 [38,42]. Conversely, 3-HBA, lacking a substituent at the 5-position, served as an agonist for both HCAR1 and HCAR2. For many HCAR2 subtype-specific agonists, such as acipimox, MK-1903, acifran, GSK256073, and SCH900271, they all contained a small or large hydrophobic group that inserted into the OBP3 region of HCAR2. In contrast, the OBP3 entrance of HCAR1 was occluded by the side chain of R71[2.60]. Thus, significant steric hindrance precluded these HCAR2-specific agonists from binding to HCAR1. Based on

these profound structural insights, we further summarized the agonist structural features that matched the HCAR1 and HCAR2 orthosteric pocket. Overall, our structural analysis provides a comprehensive understanding of the ligand recognition, selectivity, activation, and G protein coupling mechanism of HCAR1, which is important for the rational design of novel therapeutic drugs targeting HCAR1.

## Methods

### Expression and purification of the HCAR1-Gi1 and HCAR2-Gi1 complexes

Wild-type human HCAR1 and HCAR2 were cloned into the pFastBac vector (Gibco) respectively, incorporating an N-terminal hemagglutinin (HA) signal sequence, Flag tag, and HRV 3C protease site, as well as a C-terminal His tag. Dominant-negative Gαi1 (DNGαi1) with mutations (G203A and A326S) was constructed in the same manner as HCAR1 and HCAR2. The Gβ1γ2 expression vector was created using the pFastBac Dual vector (Gibco). Notably, to enhance the stability of the HCAR1-Gi1 complexes, the NanoBiT tethering strategy was applied by fusing a LgBiT subunit (Promega) at the HCAR1 C-terminus, and a SmBiT subunit (peptide, VTGYRLFEEIL) at the C-terminus of Gβ. HCAR, DNGαi1, and Gβ1γ2 proteins were co-expressed in *Spodoptera frugiperda* Sf9 cells (Invitrogen) using the Bac-to-Bac baculovirus expression system. Cells were cultivated in suspension to a density of $4 \times 10^6$ cells mL$^{-1}$ at 27 °C and infected with virus at a ratio of 10:10:1 (HCAR: DNGαi1: Gβ1γ2). Cells were harvested after 48 h of infection by centrifugation and were stored at −80 °C.

To generate the HCAR1-Gi1 and HCAR2-Gi1 complexes, cell pellets were thawed and suspended in lysis buffer [10 mM HEPES (pH 7.5), 0.5 mM EDTA] supplemented with 50 μM CHBA (MCE HY-W016868), MK1903 (MCE HY-107581), SCH-900271 (MCE HY-111143) or without an agonist. Samples were rotated at 4 °C for 1 h to induce the formation of HCAR1-Gi1 and HCAR2-Gi1 complexes. A Dounce homogenizer was used to homogenize and collect cell membranes in a solubilization buffer [20 mM HEPES (pH 7.5), 100 mM NaCl, 50 μM agonist, 10% glycerol, 1% (w/v) n-Dodecyl-b-D-maltoside (DDM), 0.1% (w/v) cholesteryl hemisuccinate (CHS), 0.2 μg mL$^{-1}$ leupeptin, 100 μg mL$^{-1}$ benzamidine, 10 mM MgCl$_2$, 5 mM CaCl$_2$, 1 mM MnCl$_2$, 100 μU mL$^{-1}$ lambda phosphatase (NEB), and 25 μU mL$^{-1}$ apyrase]. After a 2-h incubation at 4 °C, the supernatant was centrifuged and then incubated with anti-Flag M1 antibody affinity resin at 4 °C for 1 h. The M1 resin was washed with wash buffer [20 mM HEPES (pH 7.5), 100 mM NaCl, 50 μM agonist, 0.1% DDM, 0.01% CHS, and 2 mM CaCl$_2$]. The buffer underwent a stepwise transition from DDM to lauryl maltose neopentyl glycol (LMNG). Then, the M1 resin was washed with LMNG buffer [20 mM HEPES (pH 7.5), 100 mM NaCl, 50 μM agonist, 0.01% (w/v) LMNG, 0.001% CHS, and 2 mM CaCl$_2$]. The complex was eluted with elution buffer [20 mM HEPES (pH 7.5), 100 mM NaCl, 50 μM agonist, 0.00075% LMNG, 0.00025% (w/v) glycol-diosgenin (GDN), 0.0001% CHS, 5 mM EDTA, and 200 μg mL$^{-1}$ synthesized Flag peptide]. The eluted protein was incubated for 2 h on ice, with the antibody fragment scFv16 at a molar ratio of 1:1.5 [54]. A pre-equilibrated Superdex 200 Increase 10/300 column (GE Healthcare) with buffer [20 mM HEPES (pH 7.5), 100 mM NaCl, 0.00075% LMNG, 0.00025% GDN, 0.0001% CHS, and 50 μM agonist] was used to further purify the complex. The ultrafiltration tube was used to pure HCAR-Gi1-scFv16 complex, and the product was flash-frozen in liquid nitrogen until further use.

### Cryo-grid preparation and EM data collection

The pre-discharged 100 Holey Carbon film (Au, 300 mesh, N1-C14nAu30-01) with a Tergeo-EM plasma cleaner was used for preparing the cryo-EM sample. Subsequently, 3 μL of the purified HCAR-Gi1-scFv16 complex was applied to the grid. The sample was incubated for 3 s and blotted for 2 s using the freezing plunger Vitrobot I (Thermo Fisher Scientific, USA) under 10 °C and 100% humidity. Grids were quickly frozen in liquid ethane cooled by liquid nitrogen and stored in liquid nitrogen until checked. The 300-kV Titan Krios Gi3 microscope (Thermo Fisher Scientific FEI, the Kobillka Cryo-EM Center of the Chinese University of Hong Kong, Shenzhen) was used to inspect the grids and capture cryo-EM data of

the HCAR-Gi1-scFv16 complex. Movies were recorded using the Gatan K3 BioQuantum camera at a magnification of 105,000, with a pixel size ranging from 0.83 to 0.85 Å. The GIF-quantum energy filter (Gatan, USA) was utilized to eliminate inelastically scattered electrons, with a slit width set to 20 eV. The movie stacks were automatically acquired with a defocus range from −1.1 to −2.0 μm. The exposure time was 2.5 s, and frames were collected for a total of 50 frames (0.05 s/frame) per sample. The dose rate was 21.2 e/pixel/s. Semiautomatic data acquisition was performed using SerialEM 3.7.

## Image processing and 3D reconstructions

The image processing strategy followed a hierarchical approach as described in a previously published method [55]. Briefly, data binned by 4 times was used for micrograph screening and particle picking. The data with 2-time binning was used for particle screening and classification. Following the initial cleaning, particles were extracted from the original clean micrograph, and the resulting dataset underwent final cleaning and reconstruction. Raw movie frames were aligned using MotionCor2 with a 9 × 7 patch [56], and contrast transfer function (CTF) parameters were estimated using Gctf and ctf in JSPR [57]. Only micrographs with consistent CTF values, including defocus and astigmatism, were retained for subsequent image processing.

For the HCAR1-Gi1-scFv16 protein with CHBA, 3,704 movies were processed with cryoSPARC v4.1.1 [58]. Each movie stack underwent patch motion correction, and a total of 3,616,426 particles were extracted using auto-picking. After three rounds of 2D classification, the number of good quality particles was reduced to 930,340. A further reduction to 518,325 particles was achieved through 3D classification and Ab-initio reconstruction. A 3.16 Å resolution density map at FSC 0.143 was obtained through homogeneous refinement, non-uniform refinement, and local refinement of the initial particle map.

For the HCAR1-Gi1-scFv16 protein in the apo form, 4,766 movies were processed with cryoSPARC v4.1.155. Each movie stack underwent patch motion correction, and a total of 3,697,167 particles were extracted using auto-picking. After three rounds of 2D classification, the number of good quality particles was reduced to 487,455. A further reduction to 208,323 particles was achieved through 3D classification and Ab-initio reconstruction. A 3.36 Å resolution density map at FSC 0.143 was obtained through homogeneous refinement, non-uniform refinement, and local refinement of the initial particle map.

For the HCAR2-Gi1-scFv16 protein with MK-1903, 3,996 movies were processed with cryoSPARC v4.1.155. Each movie stack underwent patch motion correction, and a total of 3,572,266 particles were extracted using auto-picking. After three rounds of 2D classification, the number of good quality particles was reduced to 516,852. A further reduction to 439,381 particles was achieved through 3D classification and Ab-initio reconstruction. A 2.68 Å resolution density map at FSC 0.143 was obtained through homogeneous refinement, non-uniform refinement, and local refinement of the initial particle map.

For the HCAR2-Gi1-scFv16 protein with SCH900271, 2,628 movies are processed with cryoSPARC v4.1.155. Each movie stack underwent patch motion correction, and a total of 2,106,512 particles were extracted using auto-picking. After three rounds of 2D classification, the number of good quality particles was reduced to 501,574. A further reduction to 277,250 particles was achieved through 3D classification and Ab-initio reconstruction. A 3.06 Å resolution density map at FSC 0.143 was obtained through homogeneous refinement, non-uniform refinement, and local refinement of the initial particle map.

## Model building and refinement

The models of MK-1903- and SCH900271-bound HCAR2 were built with the reported niacin-bound HCAR2 cryo-EM structure (PDB: 8IJA) as the template. The models of apo and CHBA-bound HCAR1 were constructed using the initial template from the AlphaFold Protein Structure Database. The Gi1-scFV16 model was constructed using the FPR2-Gi1 cryo-EM structure (PDB: 6OMM) as a template [59]. All models were docked into the density maps using UCSF Chimera.

This was followed by iterative manual adjustments and rebuilding in COOT 0.9.7, along with phenix.realspace refinement. The final refinement model statistics underwent validation by Phenix. Molecular graphics figures were generated using UCSF Chimera, UCSF ChimeraX, and PyMOL. The final refinement statistics underwent validation using Molprobity, as presented in S1 Table. Notably, a few residues observed in the disallowed regions are from the structures of CHBA-bound HCAR1 (D12 and P69) and SCH900271-bound HCAR2 (L245).

## Cyclic AMP (cAMP) assay

The Gi/o-cAMP assay was carried out using a cAMP-Gi/o kit (Cisbio, 62AM9PEB). Wild-type HCAR1, wild-type HCAR2, and their mutants were cloned into a pcDNA3.1 vector. HEK-293 cells (ATCC CRL-1573) were seeded in 24-well culture plates at a density of 70−90% cells per well before transfection. Subsequently, the cells were transiently transfected with the plasmid using Lipofectamine 3,000 reagent (Invitrogen, L3000). After 36 h, the culture medium was removed from the cells, and they were washed with PBS buffer. The transfected cells were then plated into 384-well plates (4,000 cells per well) in a stimulation buffer and treated with 20 μM forskolin, 500 μM IBMX, and the test agonist for 30 min at 37 °C. Then 5 μL of cAMP Eu-cryptate reagent and 5 μL of anti-cAMP-d2 working solution were added to the 384-well plates and incubated for 1 h [60]. Fluorescence signals were detected at 620/665 nm using the Multimode Plate Reader (PerkinElmer EnVision 2,105) [61]. Data were analyzed with GraphPad Prism 9.0. The experiments were conducted in triplicate.

## Cell surface expression testing

Flow cytometry analysis was used to determine the expression levels of the HCAR1 and HCAR2 plasmids in HEK-293 cells. The levels were then utilized for normalizing the cAMP measurement. Specifically, transfected cells were blocked with 5% BSA for 15 min at room temperature, followed by labeling with anti-flag antibody (1:100, Thermo Fisher) for 1 h at 4 °C. Following a wash with PBS buffer, the cells were incubated with anti-mouse Alexa 488-conjugated secondary antibody (1:300, Beyotime) at 4 °C in the dark for 1 h. Approximately 10,000 cellular events were then evaluated for each sample using a BDAccuri C6 Plus flow cytometer. Fluorescence intensity was quantified, and the experiment was repeated three times. Values are presented as the mean ± SEM, and data analysis was performed using GraphPad Prism 9.0.

## Molecular dynamics simulation

The membrane builder module in CHARMM-GUI server [62] was used to prepare the simulation inputs, including a membrane of pre-equilibrated (310 K) POPC lipids based on the OPM database alignment [63], TIP3P solvent with 0.15 M $Na^+$/$Cl^-$ ions, and the CHARMM36 forcefield [64]. The force field of the ligands was generated by the CGenFF program [65]. All MD simulations were performed using GROMACS-2019.4 [66]. The CHARMM 36 m forcefield was used to describe the interactions in the system. Energy minimization was performed for 5,000 steps by the steepest descent algorithm. Then a 250 ps NVT simulation was performed at 310 K for solvent equilibration, followed by a 1.6 ns NPT equilibration to 1 atm using the Parrinello–Rahman barostat [67]. All MD simulations were performed with a time-step of 2 fs. Long-range electrostatic interactions were treated by the particle-mesh Ewald method [68]. The short-range electrostatic and van der Waals interactions both used a cutoff of 10 Å. All bonds were constrained by the LINCS algorithm [69]. Here, MD simulations were started from the solved structures of the MK1903-bound HCAR2 and the CHBA-bound HCAR1. Simulation runs for 200 ns. The trajectory was analyzed by the python package MDAnalysis [70].

## Supporting information

**S1 Fig. Cryo-EM data processing of the HCAR1-Gi1 signaling complex in the CHBA-bound form. (A)**. Size exclusion chromatography profile and SDS–PAGE of the HCAR1-Gi1 complex. **(B)**. Representative micrograph of the complex particles. **(C)**. Representative 2D averages. **(D)**. Workflow for cryo-EM image processing. **(E)**. Gold-standard FSC curves of the

3D reconstructions. **(F)**. Local resolution map of the complex. **(G)**. Representative density maps and models for TM1–7 and ECL2 of HCAR1 and the $\alpha$ helices of G$\alpha$i1 ($\alpha$N and $\alpha$5). The original gel image can be found in S1 Raw Images.
(TIF)

**S2 Fig. Cryo-EM data processing of the HCAR1-Gi1 signaling complex in the apo form. (A)**. Size exclusion chromatography profile and SDS–PAGE of the HCAR1-Gi1 complex. **(B)**. Representative micrograph of the complex particles. **(C)**. Representative 2D averages. **(D)**. Workflow for cryo-EM image processing. **(E)**. Gold-standard FSC curves of the 3D reconstructions. **(F)**. Local resolution map of the complex. **(G)**. Representative density maps and models for TM1–7 and ECL2 of HCAR1 and the $\alpha$ helices of G$\alpha$i1 ($\alpha$N and $\alpha$5). The original gel image can be found in S1 Raw Images.
(TIF)

**S3 Fig. Relative expression of wild-type and mutants of HCAR1 and HCAR2.** Relative cellular expression is determined by FACS analysis. The data are presented as means ± SEM. The experiments are performed in triplicate. The underlying data can be found in S1 Data.
(TIF)

**S4 Fig. Structural comparison between HCAR1, HCAR2 (PDB: 8J6Q), and HCAR3 (PDB: 8IHJ). (A)**. Superposition of the extracellular architecture of HCAR1, HCAR2, and HCAR3. All HCAR receptors have three disulfide bonds and are divided into three groups: (1) Cys$^{N-term}$-Cys$^{ECL2}$, (2) Cys$^{45.50}$–Cys$^{3.25}$, and (3) Cys$^{N-term}$–Cys$^{ECL3}$. Groups 2 and 3 are conserved, while group 1 displays a difference in its spatial position. **(B)**. Movement of the N-terminus and ECL3 in HCAR1 relative to those in HCAR2 and HCAR3. The structures of HCAR receptors are colored differently. Royal blue, HCAR1, light gray, HCAR2; sandy brown, HCAR3; khaki, N-terminal loop; pale green, ECL1; purple, ECL2; dark gray, ECL3; magenta arrow, shift in HCAR1 with respect to HCAR2 and HCAR3.
(TIF)

**S5 Fig. Sequence alignment of HCAR1, HCAR2, and HCAR3.** Positions that are identical between the receptors are highlighted with a red background.
(TIF)

**S6 Fig. Density maps of critical residues in the apo and CHBA-bound HCAR1. (A)**. Density maps of hydrophilic and hydrophobic residues in the OBP region. **(B)**. Density maps of key activation motifs. **(C)**. Density maps of key residues in the HCAR1-Gi1 interface. Royal blue-dark magenta, CHBA-HCAR1-Gi1; forest green-beige, apo-HCAR1-Gi1; orange, CHBA.
(TIF)

**S7 Fig. Analysis of the substituent character at CHBA's 5-position. (A)**. Chemical structures of compounds substituted at CHBA's 5-position. The EC$_{50}$ values are obtained from the previous report [38]. **(B)**. Hydrophilic and hydrophobic properties of CHBA and OBP in HCAR1. **(C−E)**. Predicted binding modes of compound 5 **(C)**, 6 **(D)**, and 7 **(E)** with HCAR1. The pockets and residues are colored according to hydrophobicity (turquoise) and hydrophilicity (orchid). The red and yellow polygon represents the possible region of steric hindrance.
(TIF)

**S8 Fig. Different interaction patterns of ECL2 with the agonist in HCAR1 and HCAR2.** Conformational changes of ECL2 in the simulation trajectories of the HCAR1 **(A)** and HCAR2 **(B)** complexes at 0, 50, 100, 150, and 200 ns. RMSD of the ECL2 backbone atoms in the HCAR1 **(C, E)** and HCAR2 **(D, F)** complexes. The hydrogen bonds between ECL2 and ligand during the trajectories in HCAR1 **(G)** and HCAR2 **(H)** complexes. Royal blue, HCAR1; light gray, HCAR2. The underlying data can be found in S1 Data.
(TIF)

**S9 Fig. Analysis of OBP and ABP features in HCAR1 and HCAR2. (A)**. Binding mode comparison of CHBA in OBP. **(B)**. Effects on Gi-mediated cAMP by single point mutations of R79$^{ECL1}$W in HCAR1 and W91$^{ECL1}$R in HCAR2. The data are presented as means ± SEM. The experiments are performed in triplicate. The underlying data can be found in S1 Data. **(C)**. Binding mode comparison of compound 9*n* in ABP. Royal blue, HCAR1; light gray, HCAR2; orange, CHBA; pink, compound 9*n*. **(D)** Selectivity of many other HCAR1 agonists. The EC$_{50}$ values are obtained from the previous report [42].
(TIF)

**S10 Fig. Cryo-EM data processing of the HCAR2-Gi1 signaling complex in the MK-1903-bound form. (A)**. Size exclusion chromatography profile and SDS–PAGE of the HCAR2-Gi1 complex. **(B)**. Representative micrograph of the complex particles. **(C)**. Representative 2D averages. **(D)**. Workflow for cryo-EM image processing. **(E)**. Gold-standard FSC curves of the 3D reconstructions. **(F)**. Local resolution map of the complex. **(G)**. Representative density maps and models for TM1–7 and ECL2 of HCAR2 and the *α* helices of Gαi1 (αN and α5). The original gel image can be found in S1 Raw Images.
(TIF)

**S11 Fig. Cryo-EM data processing of the HCAR2-Gi1 signaling complex in the SCH900271-bound form. (A)**. Size exclusion chromatography profile and SDS–PAGE of the HCAR2-Gi1 complex. **(B)**. Representative micrograph of the complex particles. **(C)**. Representative 2D averages. **(D)**. Workflow for cryo-EM image processing. **(E)**. Gold-standard FSC curves of the 3D reconstructions. **(F)**. Local resolution map of the complex. **(G)**. Representative density maps and models for TM1–7 and ECL2 of HCAR2 and the *α* helices of Gαi1 (αN and α5). The original gel image can be found in S1 Raw Images.
(TIF)

**S12 Fig. Interaction analysis of MK-1903 and SCH900271 with HCAR2 in the OBP region. (A, B)**. Specific interactions of MK-1903 with HCAR2. Effects on Gi-mediated cAMP by single point mutations of key residues that interact with MK-1903. **(C, D)**. Specific interactions of SCH900271 with HCAR2. Effects on Gi-mediated cAMP by single point mutations of key residues that interact with SCH900271. The data are presented as means ± SEM. The experiments are performed in triplicate. The underlying data can be found in S1 Data. **(E, F)** Density maps of MK-1903, SCH900271, and surrounding key residues. The ligands and residues are shown using a stick representation. Pink-medium slate blue, MK-1903-HCAR2; gray-aquamarine, SCH900271-HCAR2.
(TIF)

**S13 Fig. Binding modes of HCAR2 with various subtype-specific agonists. (A−F)**. Different binding modes of HCAR2 with acipimox (PDB: 8I7V) **(A)**, acifran (PDB: 8IHI) **(B)**, MK-1903 **(C)**, SCH900271 **(D)**, GSK256073 (PDB: 8I7W) **(E)**, and MK-6892 (PDB: 8IJD) **(F)**. Except for MK-6892, all other agonists bind exclusively to the OBP region. Yellow green-slate gray, acipimox-HCAR2; purple-lemon chiffon, acifran-HCAR2; pink-medium slate blue, MK-1903-HCAR2; gray-aquamarine, SCH900271-HCAR2; rose red-light sky blue, GSK256073-HCAR2; cyan-light magenta, MK-6892-HCAR2.
(TIF)

**S14 Fig. Comparison of different ligand binding modes of HCAR1 and HCAR2 in the OBP regions. (A−E)**. Pairwise comparisons of the CHBA-HCAR1 versus acipimox-HCAR2 **(A)**, MK-1903-HCAR2 **(B)**, acifran-HCAR2 **(C)**, SCH900271-HCAR2 **(D)**, and GSK256073-HCAR2 **(E)**. Orange-royal blue, CHBA-HCAR1; yellow green-slate gray, acipimox-HCAR2; pink-medium slate blue, MK-1903-HCAR2; purple-lemon chiffon, acifran-HCAR2; gray-aquamarine, SCH900271-HCAR2; rose red-light sky blue, GSK256073-HCAR2. **(F)**. Superposition of the subtype-specific agonists of HCAR1 and HCAR2.
(TIF)

**S15 Fig. Comparison of the ECL2 regions in the apo states of HCAR1 and HCAR2. (A)**. ECL2 region in the apo state of HCAR1. **(B)**. Effects of increasing HCAR1 receptor concentration on Gi-mediated cAMP. The data are presented as means ± SEM. The experiments are performed in triplicate. The underlying data can be found in S1 Data. **(C)**. Superposition of the ECL2 regions in the apo states of HCAR1 and HCAR2. Forest green, apo-HCAR1; chocolate, apo-HCAR2. (TIF)

**S16 Fig. Comparison of the HCAR1-Gi1 with HCAR2-Gi1 and HCAR3-Gi1. (A)**. Sequence alignment of ICL2 and ICL3 regions in HCAR1–3. Positions that are identical between the receptors are highlighted with a red background. **(B)**. Superposition of the architecture of ICL2 and ICL3 regions in HCAR1–3. **(C−F)**. Detailed distribution of amino acids in the ICL2 of HCAR1–3. **(G−J)**. Detailed distribution of amino acids in the ICL3 of HCAR1–3. Royal blue-dark magenta, CHBA-HCAR1-Gi1; light gray-dark khaki, HCAR2-Gi1; sandy brown-medium aquamarine, HCAR3-Gi1. (TIF)

**S17 Fig. Structural comparison between CHBA-HCAR1, β-OHB-HCAR2 (PDB: 8J6Q), AA-FFAR2 (PDB: 8J24), and BA-FFAR3 (PDB: 8J21). (A)**. Superposition of the architecture of CHBA-HCAR1, β-OHB-HCAR2, AA-FFAR2, and BA-FFAR3. **(B)**. Detailed polar interactions of β-OHB with HCAR2. **(C)**. Detailed polar interactions of AA with FFAR2. **(D)**. Detailed polar interactions of BA with FFAR3. **(E)**. Comparison of the key amino acids responsible for ligand recognition in the OBP region of HCAR1 and HCAR2. **(F)**. Comparison of the key amino acids responsible for ligand recognition in the OBP region of FFAR2 and FFAR3. **(G)**. Sequence alignment of key amino acids in HCAR1/2 and FFAR2/3. Orange-royal blue, CHBA-HCAR1; medium purple-light gray, β-OHB-HCAR2; dark blue-dark khaki, AA-FFAR2; dark green-thistle, BA-FFAR3. (TIF)

**S1 Table. Cryo-EM data collection, refinement, and validation statistics.** (TIF)

**S1 Raw Images. Uncropped Coomassie-stained SDS–PAGE gel used for S1A, S2A, S10A and S11A Figs.** (PDF)

**S1 Data. The raw data for Figs 2F–2H, 4D, 5D, S3, S9B, S12B, S12D, and S15B.** (XLS)

## Acknowledgements

We would like to thank the Kobilka Cryo-Electron Microscopy Center in the Chinese University of Hong Kong, Shenzhen, for cryo-electron microscopy analysis, and their technicians for kind help and technical support. We would like to thank the Warshel Institute for Computational Biology (funding from Shenzhen City and Longgang District) for computational work.

## Author contributions

**Conceptualization:** Xin Pan, Fang Ye, Peiruo Ning, Yang Du.

**Data curation:** Xin Pan, Fang Ye, Peiruo Ning, Yiping Yu.

**Funding acquisition:** Xin Pan, Kaizheng Gong, Yang Du.

**Investigation:** Xin Pan, Fang Ye, Peiruo Ning, Yiping Yu, Zhiyi Zhang, Jingxuan Wang, Geng Chen, Zhangsong Wu, Chen Qiu, Jiancheng Li, Bangning Chen.

**Methodology:** Xin Pan, Fang Ye, Peiruo Ning, Yiping Yu.

**Supervision:** Lizhe Zhu, Chungen Qian, Kaizheng Gong, Yang Du.

**Validation:** Xin Pan, Peiruo Ning, Yang Du.

**Visualization:** Xin Pan, Peiruo Ning, Yang Du.

**Writing – original draft:** Xin Pan, Peiruo Ning, Yang Du.

**Writing – review & editing:** Xin Pan, Peiruo Ning, Kaizheng Gong, Yang Du.

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
