## [Editor Report · Decision Letter 0]

27 Oct 2024

Dear Dr Du,

Thank you for submitting your manuscript entitled "Molecular basis for ligand recognition and selectivity of human lactate receptor HCAR1" for consideration as a Research Article by PLOS Biology. Please accept my sincere apologies for the delay in getting back to you as we consulted with an academic editor about your submission.

Your manuscript has now been evaluated by the PLOS Biology editorial staff, as well as by an academic editor with relevant expertise, and I am writing to let you know that we would like to send your submission out for external peer review.

Once your full submission is complete, your paper will undergo a series of checks in preparation for peer review. After your manuscript has passed the checks it will be sent out for review. To provide the metadata for your submission, please Login to Editorial Manager (https://www.editorialmanager.com/pbiology) within two working days, i.e. by Oct 29 2024 11:59PM.

Kind regards,

Richard

Richard Hodge, PhD

rhodge@plos.org

PLOS

---

## [Decision Letter · Decision Letter 1]

19 Dec 2024

Dear Dr Du,

Thank you for your continued patience while your manuscript "Molecular basis for ligand recognition and selectivity of human lactate receptor HCAR1" was peer-reviewed at PLOS Biology. Please accept my sincere apologies for the delays that you have experienced during the peer review process. Your manuscript has now been evaluated by the PLOS Biology editors, an Academic Editor with relevant expertise, and by three independent reviewers.

In light of the reviews, which you will find at the end of this email, we would like to invite you to revise the work to thoroughly address the reviewers' reports.

As you will see below, the reviewers are generally positive about your manuscript and think it is well-written and presented. Reviewer #2 would like to see additional MD simulation data included to explore the conformational dynamics of the receptors when bound to agonists, as well as directly demonstrating the basal activity of HCAR1. The reviewers also suggest improvements to the figure presentation and textual revisions to the discussion section to contextualize the findings more broadly within the field.

Given the extent of revision needed, we cannot make a decision about publication until we have seen the revised manuscript and your response to the reviewers' comments. Your revised manuscript is likely to be sent for further evaluation by all or a subset of the reviewers.

**IMPORTANT - SUBMITTING YOUR REVISION**

*Re-submission Checklist*

*Published Peer Review*

*PLOS Data Policy*

*Blot and Gel Data Policy*

Best regards,

Richard

Richard Hodge, PhD

rhodge@plos.org

REVIEWS:

Reviewer #1: Pan et al. performed functional analysis based on structural analysis using electron microscopy of three agonist compounds (CHBA, MK-1903, SCH900271) for HCAR1 and HCAR2 GPCRs. In particular, the correlation analysis of mechanism and structure based on the detailed structure of HCAR1 and the mutant experiments using the CHBA-bound form is new information. However, as the structure of HCAR2 has already been reported by four groups (Nat Commun 15:5364, 2024; Nat Commun 14:7620, 2023; Nat Commun 14:7150, 2024; Nat Commun 14:5899, 2023), the importance of this paper is slightly reduced, but the structural information provided by the new compounds is considered to be of sufficient value.

Minor comments.

1. There are disordered amino acid residues in the structures of HCAR1-CHBA and HCAR2-SCH900271 in Supplementary Table S1, and the names of these residues should be clearly stated in the main text.

2. In 'Model building and refinement' you say that the structures of HCAR1 and 2 were analyzed using the predicted structure from AlphaFold as the initial model. There must be structures already analyzed.

Reviewer #2 (Cheng Zhang, signs review): In this manuscript, Dr. Yang Du and his colleagues present cryo-EM structures of HCAR1 and HCAR2 in complex with Gi. The study addresses the molecular basis of ligand recognition and selectivity for the HCA receptors, a topic of high importance for therapeutic applications in cancer and metabolic disorders. The use of cryo-EM and mutagenesis studies provides a comprehensive structural analysis of HCAR1 and 2 and their interactions with ligands. The identification of key residues and structural features that influence ligand specificity is critical for drug development. The manuscript is well-structured with detailed figures and descriptions.

Findings could significantly advance the design of subtype-specific drugs targeting HCAR1 and HCAR2.

Comments:

1. Some of the ligands, such as CHBA and MK1903, are relatively small. The cryo-EM density alone does not allow for unambiguous modeling based solely on the density map. How did the authors ensure that the orientations of these ligands are correct? Is it possible for both ligands to be positioned in the opposite direction?

2. The cryo-EM density of critical residues, particularly those examined in the mutagenesis studies, should be shown to provide evidence for the accuracy of the structural modeling.

3. The authors state that HCAR1 exhibits high basal activity based on the fact that they could assemble the ligand-free complex with Gi in their structural studies. However, this is insufficient. Experimental data are required to directly demonstrate the basal activity of HCAR1.

4. The authors suggest that the different conformations of ECL2 in HCAR1 and HCAR2 partly account for ligand selectivity based on their structural comparison analysis. However, these differences might be artifacts of cryo-EM modeling, considering the low resolution of the receptor extracellular regions. Conducting additional MD simulations would be valuable to explore the conformational dynamics of ECL2 in these two receptors when bound to subtype-selective agonists.

5. Some of the content in the Introduction and Discussion regarding the functional roles of HCAR1 is redundant. Streamlining the Discussion section is recommended.

6. It could be informative to compare the structures of HCAR1 and HCAR2 with those of short-chain fatty acid receptors FFA2 and FFA3, as their endogenous ligands share a high degree of chemical similarity.

Reviewer #3: The manuscript by Pan et al, "Molecular basis for ligand recognition and selectivity of human lactate receptor HCAR1", describes several cryo-EM structures of HCAR receptors (HCAR1 and HCAR2) bound to Gi heterotrimer, and specific compounds: apo and CHBA-bound (HCAR1), MK-1903- and SCH900271-bound (HCAR2). The authors analyse the functional effects of the mutations on receptor activation and analyse the determinants of receptor specificity based on their structures.

The manuscript is well written, and the illustrations are of good quality. I have a few comments that can hopefully help the authors in revising their manuscript.

1. Figure 1. It is understandable that the GPCR field is very much used to cryo-EM structures and without any labelling one can recognise the receptor and the G protein - however to a reader outside of the field it would be helpful to see which part is in the membrane, which part is peripheral to the membrane, etc.

2. Is the ligand binding site in the apo HCAR1 occupied by any density?

3. Figure 4c-d. The figure panels seem to be very busy and thus difficult to understand. It could be that this impression comes from the multiple labels crammed into the figure panels. One possible way to deal with this could be to offload the compound colours to panel 4a (e.g., write compound names in the colour that is used in the c and d). But apart from this, there are many elements shown in these multiple comparison panels (c-d), which makes this figure rather difficult to follow. Some simplification would be useful.

The supplementary figure 13 features similar comparisons, but there the presentation is easier on the eye.

4. Figure 4f. Here the colours are partially overlapping with c-d, but now describing properties of the atoms. This can be somewhat confusing to the reader.

5. Figure 6 and the corresponding description. It is not clear that this part of the manuscript advances our understanding of Gi coupling. The authors just describe the interfacial interactions, stating the obvious facts. My impression is that it could be a figure that could be moved to the supplementary.

6. A figure that describes the key findings and mechanistic insights that the authors could derive from their analysis is currently missing. This would be a very useful illustration that would visually outline the main messages of this manuscript. I think such an illustration would allow the authors to end the manuscript with a powerful statement (if they manage to do it). Such a figure could work well with a revised Discussion.

7. I would recommend to revise the Discussion by removing the redundant paragraph 1 (or fusing it with the introduction) - and adding the figure (see point 5 above) that would help visualise the key findings mentioned in the discussion. The Discussion part could also be a good opportunity to look more broadly at the findings in the context of the whole GPCR/Gi field, not just repeating what was already said in the results and not focusing exclusively on HCARs.

8. Supplementary figure 6 should show the densitiy of the ligand (CHBA), not only the residues. Similar to supplementary figures 11e-f.

---

## [Decision Letter · Decision Letter 2]

7 Mar 2025

Dear Dr Du,

Thank you for your patience while we considered your revised manuscript "Molecular basis for ligand recognition and selectivity of human lactate receptor HCAR1" for publication as a Research Article at PLOS Biology. This revised version of your manuscript has been evaluated by the PLOS Biology editors, the Academic Editor and two of the the original reviewers.

Based on the reviews, I am pleased to say we are likely to accept this manuscript for publication, provided you satisfactorily address the following data and other policy-related requests that I have provided below (A-D):

(A) We routinely suggest changes to titles to ensure maximum accessibility for a broad, non-specialist readership. In this case, we would suggest a minor edit to the title, as follows. Please ensure you change both the manuscript file and the online submission system, as they need to match for final acceptance:

“Structures of G-protein coupled receptor HCAR1 in complex with Gi1 protein reveal the mechanistic basis for ligand recognition and agonist selectivity”

(B) Thank you for providing the structural data in the PDB and EMDB databases. However, we note that the data is currently on hold for release. We ask that you please make the structures publicly available at this stage before publication.

(C) Please also ensure that each of the relevant figure legends in your manuscript include information on *WHERE THE UNDERLYING DATA CAN BE FOUND*, and ensure your supplemental data file/s has a legend.

(D) Per journal policy, if you have generated any custom code during the course of this investigation, please make it available without restrictions. Please ensure that the code is sufficiently well documented and reusable, and that your Data Statement in the Editorial Manager submission system accurately describes where your code can be found.

We expect to receive your revised manuscript within two weeks.

*Published Peer Review History*

*Press*

Best regards,

Richard

Richard Hodge, PhD

rhodge@plos.org

Reviewer remarks:

Reviewer #2 (Cheng Zhang, signs review): The authors have addressed all of my concerns.

Reviewer #3: The authors have addressed all concerns in a satisfactory manner. The only minor point that I could point to still is the quality of the new Figure 7 - I suspect the authors would be able to improve the figure quite a bit still just by making sure all of the letters / numbers are within the shapes, and the shapes are well aligned. But I would leave the necessity of such changes to the authors discretion. I think it is a great manuscript that in principle should acceptable for publication at this stage.

---

## [Editor Report · Decision Letter 3]

23 Mar 2025

Dear Yang,

On behalf of my colleagues and the Academic Editor, Raimund Dutzler, I am pleased to say that we can accept your manuscript for publication, provided you address any remaining formatting and reporting issues. These will be detailed in an email you should receive within 2-3 business days from our colleagues in the journal operations team; no action is required from you until then. Please note that we will not be able to formally accept your manuscript and schedule it for publication until you have completed any requested changes.

PRESS

Best wishes, 

Richard

Richard Hodge, PhD

rhodge@plos.org

PLOS
